# Enhanced CH₄ emissions from global wildfires likely due to undetected small fires

Junri Zhao [1,2], Philippe Ciais [1,3], Frederic Chevallier [3], Josep G. Canadell [4], Ivar R. van der Velde[5,6], Emilio Chuvieco [7], Yang Chen [8], Qiang Zhang [9], Kebin He [2,10] & Bo Zheng [1,2] ✉

Monitoring methane (CH₄) emissions from terrestrial ecosystems is essential for assessing the relative contributions of natural and anthropogenic factors leading to climate change and shaping global climate goals. Fires are a significant source of atmospheric CH₄, with the increasing frequency of megafires amplifying their impact. Global fire emissions exhibit large spatiotemporal variations, making the magnitude and dynamics difficult to characterize accurately. In this study, we reconstruct global fire CH₄ emissions by integrating satellite carbon monoxide (CO)-based atmospheric inversion with well-constrained fire CH₄ to CO emission ratio maps. Here we show that global fire CH₄ emissions averaged 24.0 (17.7–30.4) Tg yr⁻¹ from 2003 to 2020, approximately 27% higher (equivalent to 5.1 Tg yr⁻¹) than average estimates from four widely used fire emission models. This discrepancy likely stems from undetected small fires and underrepresented emission intensities in coarse-resolution data. Our study highlights the value of atmospheric inversion based on fire tracers like CO to track fire-carbon-climate feedback.

Methane (CH₄) has greater global warming potential but a shorter atmospheric lifetime than carbon dioxide (CO₂), making CH₄ emission reduction an attractive strategy to limit near-term temperature rise[1]. As the second-largest sink of hydroxyl radicals, which govern atmospheric oxidation capacity and thus control the lifetimes of many reactive species in the troposphere, CH₄ plays a crucial role in atmospheric chemistry[2]. Accurate and in-depth knowledge of the global CH₄ budget, including its source–sink patterns and spatiotemporal trends, variations, and drivers, are needed to manage CH₄ emissions in the face of climate and air pollution challenges.

Global CH₄ sources are associated with biogenic, thermogenic, or pyrogenic processes, with fires being the largest source of pyrogenic CH₄ emissions. Fires generate CH₄ emissions through incomplete combustion of biomass and soil organic carbon under hot, dry weather and high fuel load conditions. Fires are estimated to account for approximately 4% of global total CH₄ emissions (biogenic, thermogenic, and pyrogenic sources) per year[3]; however, the interannual variations in fire CH₄ emissions are comparable to those of wetlands[4]—the largest single CH₄ nature source. Moreover, CH₄ emissions generated by fires are isotopically heavier than methane of biogenic origin[5], as methane from burning C3 vegetation (e.g., trees), and especially from burning C4 vegetation (e.g. many tropical grasses, maize, sugar cane)[6], is typically enriched in ¹³C compared to biogenic emissions from wetlands, ruminants, or waste. Thus, accurate

[1]Shenzhen Key Laboratory of Ecological Remediation and Carbon Sequestration, Institute of Environment and Ecology, Tsinghua Shenzhen International Graduate School, Tsinghua University, Shenzhen, China. [2]State Environmental Protection Key Laboratory of Sources and Control of Air Pollution Complex, Beijing, China. [3]Laboratoire des Sciences du Climat et de l'Environnement, LSCE/IPSL, CEA-CNRS-UVSQ, Université Paris-Saclay, Gif-sur-Yvette, France. [4]CSIRO Environment, Canberra, ACT, Australia. [5]SRON Netherlands Institute for Space Research, Leiden, The Netherlands. [6]Department of Earth Sciences, Vrije Universiteit, Amsterdam, The Netherlands. [7]Universidad de Alcalá, Environmental Remote Sensing Research Group, Department of Geology, Geography, and the Environment, Alcalá de Henares, Spain. [8]Department of Earth System Science, University of California, Irvine, Irvine, CA, USA. [9]Ministry of Education Key Laboratory for Earth System Modeling, Department of Earth System Science, Tsinghua University, Beijing, China. [10]State Key Joint Laboratory of Environment Simulation and Pollution Control, School of Environment, Tsinghua University, Beijing, China. ✉e-mail: bozheng@sz.tsinghua.edu.cn

estimation of fire emissions is important for tracking the total and isotopic budgets of atmospheric $CH_4$. Multiple global fire emission models have been developed[7–10]; however, their discrepancies indicate the existence of large uncertainties in the calculation of fire emissions. Intercomparison studies have demonstrated a range of 6.4–13.2 (min−max) Pg $CO_2$ yr$^{-1}$ in global fire emissions across different models[11], while estimates of fire particle emissions need to be increased by 2–3 times[12,13] to align chemical transport model simulations with measured aerosol optical depth. Although few studies have evaluated fire $CH_4$ emissions, the large uncertainties in estimated emissions of other species indicate potential uncertainties in our current understanding of the global fire $CH_4$ budget.

Improving fire emission estimation is critically needed but substantially challenging due to our limited capacity to appropriately represent fire combustion conditions and characteristics. For example, images obtained from global-coverage satellites utilized to detect burned areas typically have a spatial resolution of several hundred meters[14], implying a systematic underestimation bias due to undetected small fires, especially over the tropics[15–18]. Moreover, high-intensity fires burn litter and organic horizons of soil, which poses challenges[19] for remote sensing detection and accurate estimation of fuel consumption. Further, peat burning from smoldering processes occurs in natural and disturbed peat in the Arctic and tropics, which is extremely difficult to detect via burned area observations. Burning efficiency is affected by flaming and smoldering combustion, which vary dynamically in space and time due to environmental factors[20].

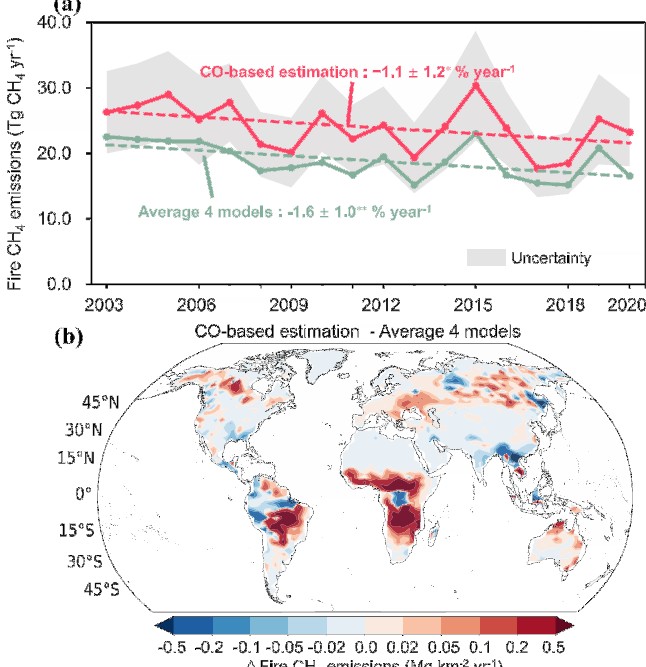

**Fig. 1 | Comparison between global CO-based fire $CH_4$ emission estimates and global fire emission model results. a** Annual trends in fire $CH_4$ emissions from CO-based estimates (red curve) and average estimates of 4 global fire emission models (green curve) from 2003 to 2020, including the fitted linear trends (dashed red and green lines). The shaded gray region represents the range of error derived from uncertainties in $CH_4$/CO emission ratios and inversion-based CO estimates, which vary with changes in dry matter and CO estimates (Methods). Trend assessments are conducted using the nonparametric Mann–Kendall test and Theil–Sen estimator, with 2003–2020 trends and uncertainties provided. Significant trends are denoted by asterisks (*$p < 0.1$ and **$p < 0.05$). **b** Spatial distribution of differences between CO-based $CH_4$ emission estimates and those of 4 fire emission models. Data averaged between 2003 and 2020 are at the spatial resolution of 3.75° longitude × 1.9° latitude.

Nevertheless, current fire emission models tend to utilize static, biome-averaged emission factors, raising the question of whether such average data are sufficiently representative[21]. These limitations hinder our understanding of fire $CH_4$ emissions and their impacts on the total $CH_4$ budget.

Inverse modeling provides a promising approach to infer $CH_4$ fluxes from ambient $CH_4$ observations[3,22] based on in situ or satellite observations. However, accurately distinguishing fire $CH_4$ emissions from total $CH_4$ fluxes is challenging due to contributions from other human and natural sources and interactions among multiple $CH_4$ sources. The measurement of carbon monoxide (CO), a well-observed tracer of fire smoke plumes, provides an alternative top-down constraint on fire emissions compared to $CH_4$ or $CO_2$ measurements alone[23,24]. CO has a short lifetime of 1 to 2 months; therefore, its ambient concentrations exhibit large deviations and gradients from background levels, which are distinct between fires and fossil fuel emissions in terms of seasonality and location. We previously developed a method to infer fire $CO_2$ emissions from satellite retrieval of CO[25,26]. Reconstruction of burning efficiency maps, which represent the fraction of carbon emitted as $CO_2$ during burning, based on satellite-observed CO inversion lays the foundation for linking combustion conditions and fire-carbon release. This approach was developed, maintained, and has been successfully applied by our research group to estimate global fire CO and $CO_2$ emission trends and drivers for the past two decades[25,27].

Herein, we explore the possibility of constraining fire $CH_4$ emissions based on fire CO inversion, considering that both gases result from incomplete combustion of biomass. We developed new conversion functions to relate fire $CH_4$ and CO emissions by biome based on 148 sets of field measurements sampled on the ground and applied these functions post-inversion to the CO emissions from fires derived from our inversion system. Field campaigns that sampled fire plumes by aircraft were further selected to evaluate the correlation between $CH_4$ and CO emissions. We compared our CO-based fire $CH_4$ emission estimates with those obtained from existing global fire emission models based on satellite-observed burned areas and fire radiative power. Differences in the results were analyzed by region and by month, and we investigated the reasons for discrepancies with previous estimates with the help of high-resolution burned areas, fire fuel consumption estimates, and land cover maps to deepen our understanding of the uncertainties associated with the global fire $CH_4$ budget.

## Results

### Global fire $CH_4$ emissions inferred from CO inversion

Our satellite-observed CO-based estimates indicated that global fires released an average of 24.0 Tg $CH_4$ yr$^{-1}$ between 2003 and 2020, which was 5.1 Tg higher than the average estimates obtained from 4 global fire emission models, including the Global Fire Emissions Database (GFED) v4.1s[10], Fire Inventory from NCAR (FINN) v2.5[28], Global Fire Assimilation System (GFAS) v1.2[8], and Quick Fire Emissions Dataset (QFED) v2.5r1[9]. The construction of these models is based on satellite-observed burned areas or fire radiative power and they are widely utilized in fire emission assessment. The average estimates of these models lay close to the lower boundary of our emission uncertainty range, considering the uncertainties in calculating the fire CO to $CH_4$ emission ratio (CO/$CH_4$ ER) (Fig. 1a). Except for FINNv2.5, the global total $CH_4$ emission estimates of the other 3 fire emission models were approximately 5–9 Tg $CH_4$ lower than our CO-based results over the 18-year period (Supplementary Table 1). FINNv2.5 employs an aggregation algorithm for burned area determination, which combines multiple detections to identify larger burned areas using satellite active fire products at a nominal 1 km$^2$ resolution, resulting in an approximately doubled estimate compared to its precedent version, FINN v1.5[7]. Compared to our fire $CH_4$ estimates, FINNv2.5 is generally

higher (Supplementary Fig. 1a), particularly from 2003–2010, though the estimates converge more closely from 2011–2020 (with an average difference of -1.5 Tg yr$^{-1}$). Spatially, our fire CH$_4$ estimates are higher than those from FINNv2.5 in most regions (Supplementary Fig. 1b), except in high-biomass areas such as the Amazon Basin and Central Africa, where Wiedinmyer et al.[28] suggested that FINNv2.5 likely overestimates emissions, as indicated by comparisons between model results and satellite observations.

The trends of our emission results were broadly consistent with those of previous emission models, albeit indicating a slightly modest downward trend. Our results revealed a slight decline in global fire CH$_4$ emissions of $-1.1\% \pm 1.2\%$ yr$^{-1}$ from 2003 to 2020 (nonparametric Mann–Kendall test, 95% confidence interval), equivalent to $-0.29 \pm 0.31$ Tg CH$_4$ yr$^{-1}$ (red curve in Fig. 1a). The average of the 4 global fire emission models exhibited a decreasing trend of $-1.6\% \pm 1.0\%$ yr$^{-1}$ ($-0.35 \pm 0.22$ Tg CH$_4$ yr$^{-1}$) during the same period (green curve in Fig. 1a). Worden et al.[23] estimated a decrease of $3.7 \pm 1.4$ Tg CH$_4$ yr$^{-1}$ in global annual average fire CH$_4$ emissions from 2001–2007 to 2008–2014 based on atmospheric inversion of satellite-based CO observations. During a similar period, our study estimated a decrease in the annual mean fire CH$_4$ emissions of $4.3 \pm 1.0$ Tg CH$_4$ yr$^{-1}$ from 2003–2007 to 2008–2014, which was similar to that estimated by Worden et al.[23] and 64% higher than that of the GFED v4.1 s model.

## Spatiotemporal distribution of fire CH$_4$ emissions

Our CO-based fire CH$_4$ emissions revealed a distinct dipole distribution pattern across latitude bands, characterized by CH$_4$ emission hotspots concentrated over tropical and boreal regions, consistent with the average estimates of the 4 global models (Supplementary Fig. 2). The higher fire CH$_4$ emissions derived from our atmospheric CO inversion compared to those from the 4 fire emission models were predominantly concentrated within the tropical latitude band spanning 30°S–15°N (Fig. 1b), accounting for the majority of the disparity in global total CH$_4$ emissions (5.7 Tg CH$_4$). By contrast, our results indicated slightly lower fire CH$_4$ emissions than those from the 4 fire emission models within the 15°N–45° N latitude band, for a total mean difference of −0.9 Tg CH$_4$ emissions. The decadal decrease in fire CH$_4$ emissions from 2003–2011 to 2012–2020 revealed by our CO-based results was attributed to the pronounced decrease in fire emissions over the 30°S–15°N latitude band (Supplementary Fig. 3), in which satellites detected a decline in grassland burning due to population growth and agriculture expansion[29], which are likely the main contributing factors.

Additionally, our CO-based results demonstrated a prominent decreasing trend in fire CH$_4$ emissions over South America, which accounted for approximately 62% of the total decadal decrease in the 30°S–0° region, a finding that was consistent with the observations by van Wees et al.[30] of reduced fire contributions due to Amazon forest loss. For the 0°–30°N region, most of the CH$_4$ emission decrease was observed in Africa, consistent with the findings of previous research[31]. However, boreal fire emissions increased since 2003 likely driven by changes in the moisture balance as the Arctic continues to warm, which partly offset the rapid decrease in tropical fire emissions and caused a gradual shift in fire emission distribution toward northern high latitudes. Our CO-based results depicted more substantial changes in tropical and boreal fire emissions since 2003 compared to those from the 4 global emission models (Supplementary Fig. 3).

During fire season months, our CO-based estimates of fire CH$_4$ emissions were approximately 52% higher than those of the 4 global fire emission models (as shown in Fig. 2a). Such conditions occur from November to March (CH$_4$ emission estimates of our model were 32%–59% higher than those of the 4 fire emission models) at 0°–15°N and from July to October (45–68% higher) at 30°S–0° (Fig. 2b), with the greatest disparity between model results occurring in October (68%). In addition, our atmospheric CO-based results indicate a larger allocation of fire CH$_4$ emissions to the late fire season (February, March, and November at 0°–15°N, and October at 30°S–0°). The importance of late fire season emissions was also reported in regional studies[26,32,33], reflected by late fire season peaks in satellite-observed CO, ammonia, and aerosol optical depth over fire regions compared to levels observed via atmospheric transport model simulations. Ramo et al.[16] highlighted the contribution of previously undetected small fires during the late fire season in Africa, which was consistent with our CO-based fire CH$_4$ emission estimates. Similarly, van der Velde et al.[18] found a better agreement with satellite-observed CO column concentrations when these small fires were accounted for.

## Undetected small fires explain higher CO-based CH$_4$ emissions

The fire CO emissions inferred from atmospheric CO inversion were constrained by satellite CO column retrieval and comprehensively evaluated in previous studies[25,27,34]. In particular, posterior CO emissions corrected the underestimation bias of simulated global CO concentrations compared with satellite and independent surface observations. Although potential uncertainties remain, we argue that the systematic bias of CO fluxes was removed and well constrained. Taking this into account, the discrepancies between our CO-based CH$_4$ emission estimates and those from global fire emission models could be due to: 1) CH$_4$/CO ERs being overestimated in our study, and 2) previous fire emission models underestimating parameters such as emission factors, burned area, or fuel combustion per unit of area burned, resulting in lower fire emissions. Therefore, we separately evaluated each possible factor and identified the leading factor.

We established fire CH$_4$/CO ER maps by grid cell and by month to correlate fire CO emissions with fire CH$_4$ emissions (see Methods) based on the spatiotemporal distribution of different types of fire and fire emission measurements across biomes (Supplementary Fig. 4). Such maps reflect the dynamics of combustion conditions that vary by location, time, and biome; however, such evaluations are challenging due to the lack of large-scale, direct, and independent measurements. Since the data from field campaigns on the ground were previously utilized in our modeling, we used independent airborne measurements from the Fire Influence on Regional to Global Environments and Air Quality (FIREX-AQ)[35] campaign over the United States and the Atmospheric Tomography Mission (ATom)[36] campaign near the South Atlantic equatorial region for further analysis of potential overestimation of CH$_4$/CO ERs. Although these two aircraft campaigns covered limited regions and periods, the data provided an independent basis for evaluation.

We identified wildfire plumes from FIREX-AQ and ATom data according to their measurement characteristics (see Methods, Supplementary Figs. 5–8). CH$_4$/CO ERs were determined as the slope of the regression line fitted to CH$_4$ and CO aircraft measurements within fire plumes, considering species loss during plume transport (Methods). The FIREX-AQ and ATom data provided consistent fire CH$_4$/CO ER values of 0.09 and 0.08 ppb/ppb, respectively (Supplementary Fig. 9), albeit they were performed over different biomes (e.g., temperate forest vs. savanna). Moreover, recent studies[37–39] based on aircraft measurements of regional wildfire smoke over the United States reported average CH$_4$/CO ER values of 0.08–0.10 ppb/ppb. For comparison, we extracted the CH$_4$/CO ERs utilized in our study from the multiannual average map (Supplementary Fig. 10, "Methods") corresponding to the time and location of the FIREX-AQ and ATom campaigns, yielding an average CH$_4$/CO ER value of 0.08 ppb/ppb. Additionally, aircraft measurements of near-field fire plumes over the savanna region in Senegal reported[40] CH$_4$/CO ERs of 0.04–0.05 ppb/ppb, consistent with the 0.04 ppb/ppb ER used for the savanna biome in this study. This alignment with previous aircraft measurements suggested that the uncertainties in CH$_4$/CO ER were not the dominant factor driving the discrepancies between model results, though further biome-specific testing is required to confirm this across all regions.

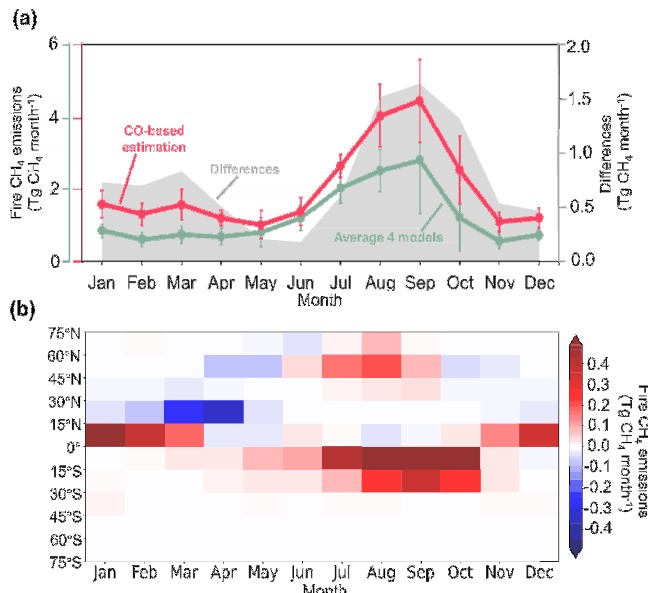

**Fig. 2 | Difference between global CO-based fire CH₄ emission estimates and global fire emission model results by month and latitude. a** Monthly fire CH₄ emissions from CO-based (red curve) estimates and average estimates of 4 fire emission models (green curve) and their difference (shaded gray region) averaged between 2003 and 2020. The error bars represent one standard deviation due to interannual variation from 2003 to 2020. **b** Differences between CO-based CH₄ emission estimates and those of four models by month and latitude.

To assess potential underestimation of burned areas, we employed the data from FireCCISFD11, a 20 m resolution data product over sub-Saharan Africa for the year 2016 which has better accuracy than existing coarse-resolution data[16], to recalculate fire CH₄ emissions using the emission intensity under the GFED v4.1 s framework (see Methods). Such CH₄ emission estimates addressed part of the underestimations in the GFED v4.1 s emission estimates over sub-Saharan Africa (Fig. 3a, b). The CO-based and FireCCISFD11-based results (GFED v4.1 s framework) were higher than those of GFED v4.1 s by 3.0 and 1.1 Tg in Northern Africa (Fig. 3a), respectively, and by 4.2 and 1.6 Tg in Southern Africa (Fig. 3b), respectively. However, the FireCCISFD11-based CH₄ estimates (GFED v4.1 s framework) remained 35%–37% lower than our CO-based results, primarily due to lower estimates over the 15°S–10°S and 5°N–10°N regions (cyan and purple curves in Fig. 3c), which accounted for 51% of the global total CH₄ emission difference despite this region having the most substantial increment in burned areas between GFED v4.1 s and FireCCISFD11, exceeding 40 Mha (depicted by the dark blue dashed curve in Fig. 3c).

We further analyzed the Sentinel-2 land cover map[16] jointly with FireCCISFD11-based burned areas. More than 40% and 50% of FireCCISFD11-based burned areas over the 15°S–10°S and 5°N–10°N regions, respectively, were covered by trees, whereas other land cover types (i.e., grassland, shrub, and cropland) dominated burned areas over other latitudinal areas in sub-Saharan Africa (Fig. 3e). We suspect that differences between the CO-based and FireCCISFD11-based CH₄ emission estimates over the 15°S–10°S and 5°N–10°N regions were probably due to the incorrect characterization of fuel load and consumption or to forest-specific emission factors, which are key components of emission intensity, in tree-dominated areas under the GFED v4.1 s framework.

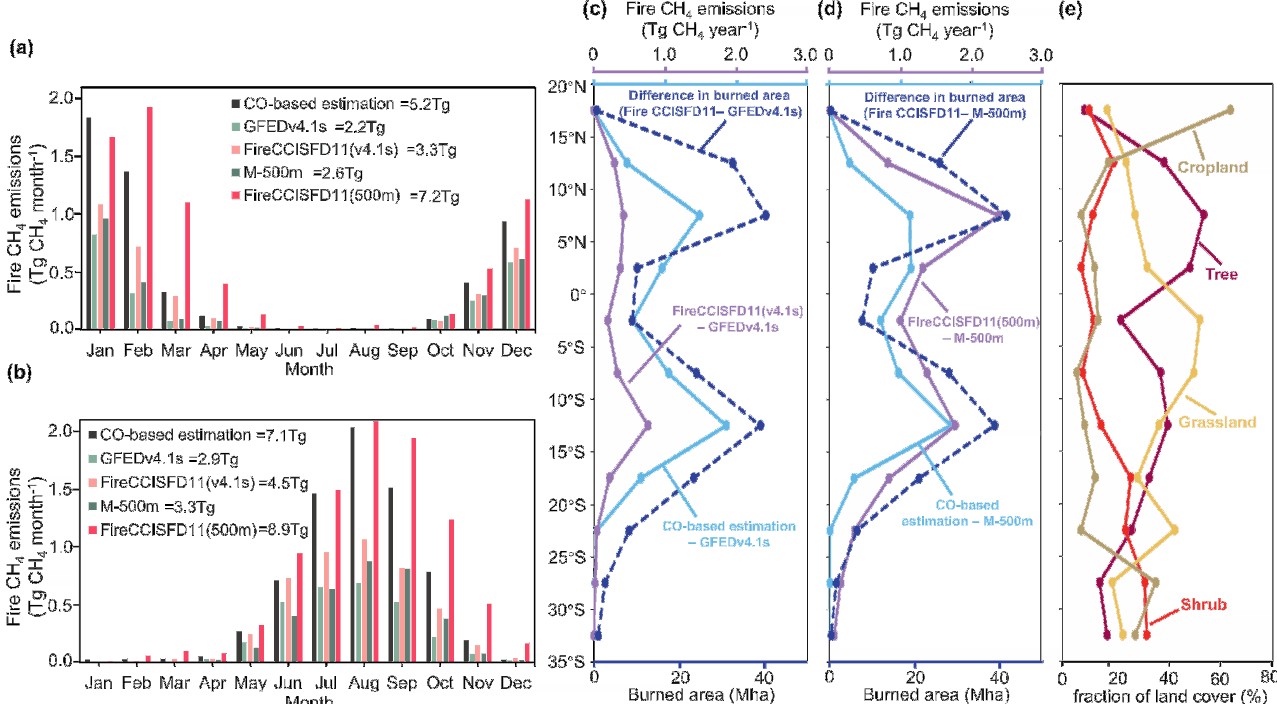

**Fig. 3 | Comparison of African fire CH₄ emissions derived from different model frameworks.** Monthly variations in fire CH₄ emission estimates based on CO estimates (black bar), FireCCISFD11-based burned areas under the M-500m (red bar) and GFED v4.1 s (pink bar) frameworks, M-500m (dark green bar), and GFED v4.1 s (light green bar) compared for the (**a**) northern and (**b**) southern hemispheres of sub-Saharan Africa in 2016. **c** Differences between fire CH₄ emission estimates based on FireCCISFD11-based burned areas (GFED v4.1 s framework) and GFED v4.1 s, CO-based and GFED v4.1 s, and burned areas FireCCISFD11 and GFED v4.1 s, shown by latitude band. **d** Differences between fire CH₄ emission estimates based on FireCCISFD11-based burned areas (M-500m framework) and M-500m, CO-based and M-500m, and burned areas FireCCISFD11 and M-500m, shown by latitude band. **e** Fractions of different land cover types over the burned area within corresponding latitudinal regions.

To evaluate this hypothesis, we further utilized the framework of the fuel consumption product[41] with a resolution of 500 m (M-500m) −but regridded to 0.25° spatial resolution to ensure commonality with GFED v4.1 s (Supplementary Fig. 11)−to recalculate fire $CH_4$ emissions with the FireCCISFD11-based burned areas for sub-Saharan Africa. The FireCCISFD11-based $CH_4$ emission estimates using the M-500m framework could resolve the disparities between the CO-based and FireCCISFD11-based $CH_4$ emission estimates (GFED v4.1 s framework), but they also surpassed our CO-based results by 1.8 and 2.0 Tg over Southern and Northern Africa, respectively. The most substantial increment between the FireCCISFD11-based (M-500m framework) and CO-based results was observed in the 5°N–10°N region (Fig. 3d), characterized by the highest tree cover fraction compared to other latitudinal bands. The aggregation of fuel consumption and burned area may have introduced errors, contributing to higher emission estimates. A recent 2019 study[18] on biomass burning found emissions for Southern Africa exceeded those of GFED v4.1 when using the unaggregated M-500m fuel consumption model and updated, region-specific dynamic emission factors[42], alongside 500 m burned area data from Sentinel-2[17]. This method estimated methane emissions at 7 Tg, which closely matches the inverse estimate from our study, albeit for a different year (Fig. 3b). Interpreting such differences is difficult, but our sensitivity analysis suggested that the fire emission models based on high-resolution burned areas or fuel data yielded larger fire $CH_4$ emission estimates than other models, which tended to move closer to our independent CO-based results than previous coarse-resolution model estimates.

## Discussion

Our study suggests that existing fire emission models may underestimate global fire $CH_4$ emissions due to their reliance on coarse-resolution burned area and emission intensity. Coarse-resolution data cannot represent the heterogeneity of fire dynamics within a coarse grid cell, and, more importantly, they are subject to large omission errors and miss small fires. Although these observations were only based on a regional analysis of Africa, this continent accounts for more than half of the global burned area and fire emissions. We acknowledge that tropical Africa is not a quasi-natural land surface, as it is heavily impacted by emissions from anthropogenic activities, which affect emission patterns and their representation in fire-related datasets. High-resolution data from different regions, especially those that consider anthropogenic activities, will aid in a more comprehensive evaluation. Emissions from small-fire types (e.g., landfill and crop residue burning) may not be accurately captured by burned-area-based products[43]. However, their plumes are likely included in satellite CO observations, which are used to constrain fire emissions in our inversion system. It should be noted that a recently developed burned area product by Chen et al.[44], employing the latest version of GFED (v5), demonstrated an approximately 61% increase in global burned area compared to GFED v4.1 s by adjusting for commission and omission errors, particularly those associated with small fires. This indicates that fire emissions based on GFED v4.1 s were largely underestimated, supporting the findings presented in this study. The upcoming fire emissions datasets based on the GFED v5-based burned areas are expected to help reconcile the discrepancies between our CO-based and GFED model $CH_4$ emission estimates.

This study is subject to potential uncertainties associated with multiple factors, mainly involving the global CO inversion and the fire $CH_4$/CO ERs developed based on field measurement data. The atmospheric CO inversion system benefits from the short atmospheric lifetime of CO and the reliability of satellite CO column retrieval. The CO inversion system was previously evaluated, demonstrating a substantial improvement in CO concentration simulation compared to independent CO observations. Regarding $CH_4$/CO ERs, evaluation against aircraft measurements revealed a close agreement between field-measured values and the data employed herein. The uncertainties persist, stemming from the inversion process, limited spatiotemporal coverage of the evaluation datasets (e.g., the FIREX-AQ and ATom campaigns were only conducted in summer and winter, respectively), and lack of peat fire plume observations. Despite these uncertainties, the lower bound of the uncertainty range (averaging 18.1 Tg yr$^{-1}$ for 2003–2020) remains in close alignment with the average estimate of four global fire emission models (18.9 Tg yr$^{-1}$ for the same period), indicating that they are unlikely to significantly affect the main conclusions of this study. Incorporating additional observational data with broad spatiotemporal coverage, such as synergistic satellite retrieval of $CH_4$ and CO column concentrations over fire regions, will improve our understanding of the dynamic changes in $CH_4$/CO ER in the future. This study presents integrated uncertainties (shaded area in Fig. 1a), which arise from the approximated uncertainties in inversion-based CO estimates and $CH_4$/CO ER uncertainties. However, we acknowledge that uncertainties remain in the emissions factor-based approach. For example, the uncertainty in hydroxyl radical (OH) concentrations− being the primary sink of CO−can significantly impact the atmospheric CO and $CH_4$ burden[4,45], highlighting the importance of accurately quantifying OH levels in fire emission estimates. The ongoing debate regarding OH variation[46] and the challenges in simulating its nonlinear chemistry in global models emphasize the need for further work to refine OH field estimates. Additionally, uncertainties in prior emission estimates and the partitioning of posterior CO fluxes across emission sectors[23] represent other key challenges in this study. These uncertainties arise from limitations in emission inventories, including outdated data and oversimplified assumptions in bottom-up calculations, which can bias modeled CO distributions and propagate through scaling factors to affect posterior estimates. Additionally, overlapping sources, such as wildfires and fossil fuels, complicate the accurate sectoral partitioning of posterior CO fluxes, potentially impacting source-specific trend analyses. Addressing these challenges requires enhanced emission inventories incorporating high-resolution observational data and the application of supplementary tracers, such as isotopic signatures or co-emitted species, to improve source attribution and refine sectoral partitioning.

Our study findings suggest that previous estimates of global fire $CH_4$ emissions based on coarse-resolution burned areas tend to be underestimated by 27%, which leads to a potentially large underestimation of global fire impacts on climate. The extent of such underestimation, based on the total difference (equivalent to 5.1 Tg yr$^{-1}$) between our results and the four models, corresponds to a significant proportion, ranging from 8% to 78%, of the total anthropogenic $CH_4$ emissions (all sectors in the EDGARv7.0[47] database) from the top 10 emitting countries (Supplementary Fig. 12). As global warming continues, wildfires are projected to occur more frequently in many parts of the world[48,49], and fire weather season will likely intensify and become longer, leading to even higher fire $CH_4$ emissions and exacerbated global warming[50]. Inadequate management of emissions from small fires (e.g., landfill, crop residue burning) in developing countries, which are exhibiting obvious growth trends in certain regions[51], leads to increased $CH_4$ emissions and the release of other harmful gases and particulates. Without improved regulation, the future may further exacerbate such emissions[52]. To enhance our understanding of fire's climate impact and support mitigation and adaptation strategies, top-down estimates of fire greenhouse gas emissions based on multiple satellites need to be integrated into a global fire monitoring and modeling system to evaluate global and regional fire-carbon budgets and resolve fire–climate feedback. Carbon and air pollution sensors are powerful tools for monitoring fire-carbon emissions directly and indirectly, respectively, with the latter being an important complement to our current fire emission monitoring system.

## Methods

### Fire CO emissions derived from atmospheric inversion

We utilized a global atmospheric inversion system to estimate global CO fire emissions from 2000 to 2020 at a spatial resolution of 3.75° longitude and 1.9° latitude. The system was developed based on the three-dimensional transport model of the Laboratoire de Météorologie Dynamique (LMDz) coupled with the Simplified Atmospheric Chemistry Assimilation System (SACS)[53,54], which has been maintained by the French Laboratoire des Sciences du Climat et de l'Environnement for the past 15 years[25,34,55,56]. This system follows Bayesian principles[57], which involves minimizing the cost function that combines prior information and satellite CO observations. These data products are connected through a global chemical transport model and weighted according to their respective uncertainties. Recent updates[25,34] to the model have enabled accurate reconstructions of the global CO budget, correcting for prior modeling biases of CO and showing good agreement with in situ CO observations. The optimized CO budgets were robust to different observational constraints, corrected misrepresentations of CO emission trends in developing countries[58], and improved the estimation of fire CO emissions during the late dry season in Africa[26].

The observational constraint in this study was the Level 2 Measurements Of Pollution In The Troposphere (MOPITT) version 9 CO column product[59], which benefits from improved cloud detection and mapping of highly polluted scenes compared with previous MOPITT retrieval versions, further enhancing the inversion system's capabilities. Prior fire emissions were obtained from GFED v4.1 s from 2003 to 2020[10]. Prior anthropogenic fossil fuel and biofuel fluxes for 2003–2019 were derived from the latest Community Emissions Data System (CEDS) emission inventory[60,61], which corrected for an overestimation of global CO emissions in the previous version. To provide prior fluxes before 2020, we utilized daily country- and sector-level $CO_2$ emission growth rates from the Carbon Monitor dataset[62,63], combined with CEDS emission data from 2019. The methodology for CO inversion is detailed in Supplementary Text 1.

### Fire CH₄ emissions derived from CO inversion

We employed our established methodology[25,26,34] to quantify fire CO emissions, and then estimated global gridded monthly fire $CH_4$ emissions based on a variation of the methodology we previously developed to reconstruct global $CO_2$ fire emissions[27], according to Eqs. (1) and (2):

$$ER_{i,j,t}^{CH_4:CO} = \frac{\sum_p DM_{i,j,t,p} \times CF_p^{CH_4:CO}}{\sum_p DM_{i,j,t,p}} \qquad (1)$$

$$E_{i,j,t}^{CH_4} = E_{i,j,t}^{CO} \times ER_{i,j,t}^{CH_4:CO} \qquad (2)$$

where $i$ and $j$ correspond to the row and column of simulation grid cells, respectively; $t$ represents an individual month between 2003 and 2020; $p$ represents the biome, including savanna, temperate forest, tropical forest, boreal forest, peatland, and agricultural land; and $ER$ represents emission ratio, $E$ represents emissions, and $CF$ signifies the conversion factor from fire CO to $CH_4$ emission factors, derived from 148 field measurements obtained from Andreae et al.[64] and other literature (Supplementary Table 3). These measurements cover a wide range of different fire types, resulting in $CF$ values ranging from 0.009 to 0.085 (g kg$^{-1}$/g kg$^{-1}$) (Supplementary Fig. 4). A statistically insignificant correlation ($p$-value < 0.05) was observed for peatland fires, and relatively lower R-values were noted for boreal forest fires. Hence, for boreal forest and peatland fires, the average $CH_4$/CO emission factor ratios determined from measurement were used as the $CF$, while for other fire types, the regression

slopes were utilized. $DM$ represents the dry matter combustion of different fire types, which is used as a weight in the calculation of gridded monthly fire $CH_4$/CO emission ratios from 2003 to 2020 (Supplementary Fig. 10), which was subsequently multiplied by the fire CO emissions derived from the LMDz-SACS inversion system to obtain CO-based fire $CH_4$ emissions (Eq. (2)).

Herein, the $DM$ data input was derived from GFED v4.1 s. Although GFED v4.1 s tends to systematically underestimate burned areas and emissions, we assumed that such errors tended to affect all ecosystem fires to a similar extent within each grid cell; therefore, the representation of spatiotemporal distribution patterns across different fire types was not subject to systematic errors. We assess the uncertainties associated with CO-based $CH_4$ emissions (shaded area in Fig. 1), according to the Eq. (3):

$$\Delta E_{i,j,t}^{CH_4} = E_{i,j,t}^{CH_4} \times \sqrt{\left(\frac{\Delta E_{i,j,t}^{CO}}{E_{i,j,t}^{CO}}\right)^2 + \left(\frac{\Delta ER_{i,j,t}^{CH_4:CO}}{ER_{i,j,t}^{CH_4:CO}}\right)^2} \qquad (3)$$

Where, $\Delta E$ and $\Delta ER$ represent the uncertainties in emissions and emission ratios, respectively. The $\Delta ER$ was derived based on Eq. (1), where we substituted the average $CH_4$/CO emission factor ratio with the standard deviation of the $CH_4$/CO emission factor ratios for $CFs$ of boreal forest and peatland, and we replaced the regression slope with the standard deviation of the residuals for other fire types. The $\Delta E$ for inversion-based CO emissions was obtained by calculating the standard deviation of monthly and grid-based CO inversion results from three sensitivity simulations in our previous study (see Table 2 in Zheng et al.[34]). Subsequently, we evaluated the uncertainty range of $\Delta E$ and $\Delta ER$, and propagated such uncertainties to estimate the emission uncertainties.

### Aircraft measurement-based evaluation of CH₄:CO emission ratio

We utilized measurements from the FIREX-AQ and ATom (https://daac.ornl.gov/ATOM/campaign/) campaigns to evaluate fire $CH_4$/CO ER.

The FIREX-AQ campaign's near real-time sampling capability enabled the detection of prominent wildfire plume signals during wildfire combustion (Supplementary Fig. 6). The measured mixing ratios of $CH_4$ and CO in these plumes reached levels as high as 3218 and 5688 ppb, respectively. The level of hydrogen cyanide (HCN), a long-lived tracer of wildfire emissions[65], reached a maximum value of 34,189 ppt. We classified data points as plumes originating from wildfires when enhanced $CH_4$, CO, and HCN levels exceeded the standard deviation of their corresponding daily mean. Since plume transport time was much shorter than CO (1 month) or $CH_4$ (9 years) lifetime[66,67], we neglected $CH_4$ and CO losses due to transport from the fire to the plume interception location during the FIREX-AQ campaign. The concentration enhancement ratio of $CH_4$ to CO was thus close to the $CH_4$/CO ER.

A total of 16 fire plume interceptions were identified in the ATom campaign. The interception data labeled #9–16 (Supplementary Table 2) were based on wildfire plume interception information reported by Chen et al.[68], which included detailed identification of fire plumes intercepted during ATom-3 and ATom-4 deployments[36] in September–October 2017 and April–May 2018, respectively. Please refer to Chen et al.[68] for further details on this dataset. As for ATom-1 (July–August) and ATom-2 (January–February) deployments, we restricted our analysis to the South Atlantic region (35°S–35°N, 65°W–10°N), which was close to the fire-prone areas of sub-Saharan Africa along the flight path.

After filtering for missing values, we obtained observation data for February 13 and 15, 2017, for which the flight tracks and time series measurements are depicted in Supplementary Figs. 5 and 7, 8, respectively. The wildfire plume signals were relatively weak due to the

relatively large distances between the plume interception locations and the fire-burning continental area. To aid in plume source attribution, we included additional measurements, such as the biomass burning (BB) fraction, which was based on the abundance of particles detected by the particle analysis by laser mass spectrometry instrument onboard the aircraft during the ATom campaign. Case-by-case identification of plumes intercepted during the Atom campaign was performed as described by Chen et al.[68], employing the following criteria: $CH_4 > 1840$ (ppb), $CO > 100$ (ppb), $HCN > 320$ (ppt), and $BB > 30\%$ for February 13, 2017, and $CH_4 > 1840$ (ppb), $CO > 120$ (ppb), $HCN > 330$ (ppt), and $BB > 50\%$ for February 15, 2017, as indicated by the dashed horizontal lines in Supplementary Figs. 7 and 8. To mitigate the influence of low-frequency fluctuations, the measurements were aggregated by averaging the observed values for CO within each 5 ppb interval ranging from 70 to 420 ppb (e.g., 70–75, 75–80,…, 415–420 ppb). Plume age was determined based on time since the most recent fire influence, which was based on back trajectories[68] obtained from ATom datasets[69], as indicated by Fire inf in Supplementary Table 2. Since the plume transport time was relatively long, we calculated first-order losses of $CH_4$ and CO and converted the concentration enhancement ratio of $CH_4$ to CO to the $CH_4$/CO ER, as described by Lutsch et al.[65]. This conversion process can be represented by the following equation:

$$ER^{CH_4:CO} = EnR^{CH_4:CO} \times \frac{\exp\left(\frac{d}{\tau_{CH_4}}\right)}{\exp\left(\frac{d}{\tau_{CO}}\right)} \qquad (4)$$

where $d$ is the age of the fire plume; $\tau$ is the atmospheric lifetime of $CH_4$ and CO (9 years and 30 days, respectively); $ER$ represents the fire $CH_4$/CO emission ratio; and $EnR$ represents the measured fire $CH_4$/CO concentration enhancement ratio.

## Other datasets used in this study
We employed the FireCCISFD11 burned area product and its corresponding land cover dataset from Sentinel-2 instruments derived from Ramo et al.[16], which covers the entire sub-Saharan Africa region at 20 m resolution for the year 2016. The FireCCISFD11 dataset was systematically validated by sampling Sentinel image pairs, and the error matrices revealed that it had substantially lower average errors in burned areas (e.g., omission and commission errors of 24.5% and 8.1%, respectively, and a Dice Coefficient of 0.83) than other global products[14,70,71]. To estimate FireCCISFD11-based fire $CH_4$ emissions (as depicted in Fig. 3), we aggregated the FireCCISFD11-based burned areas into a spatial resolution of 0.25° × 0.25° and recalculated fire emissions using emission intensity based on the GFED v4.1 s[10] and M-500 m[41] model frameworks, replacing the burned area data. The emission intensity was calculated by dividing the fire emissions by the burned areas from GFED v4.1s (0.25° × 0.25°), which is based on MODIS burned area augmented with the GFED small fires algorithm, and M-500m (0.25° × 0.25°), which is based on MODIS burned area hence the addition 'M' to M-500m, for the year 2016 (Supplementary Fig. 11a, c).

In addition, we compared our CO-based estimates with those from four global fire emission models (Supplementary Table 1), including FINNv2.5[28], QFED v2.5r1[8], GFAS v1.2[8], and GFED v4.1s[10]. We derived global anthropogenic $CH_4$ emissions for 2003–2020 from EDGARv7.0[45], as shown in Supplementary Fig. 12.

## Data availability
All of the datasets associated with the main findings of this study have been detailed in the main text and Methods section. The CO-based fire $CH_4$ emissions generated in this study can be obtained from the corresponding author upon reasonable request.

## Code availability
The code for the global atmospheric CO inversion system utilized in this study can be accessed at http://community-inversion.eu/installation.html#getting-the-code.

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

## Acknowledgements

This work was supported by the National Natural Science Foundation of China (Grant No. 42375096) and the Shenzhen Science and Technology Program (Grant No. ZDSYS20220606100806014).

## Author contributions

Conceptualization: B.Z., Methodology: B.Z. and J.Z., Investigation: B.Z. and J.Z., Visualization: J.Z., Funding acquisition: B.Z., Project administration: B.Z., Supervision: B.Z., Writing–original draft: B.Z. and J.Z., Writing–review and editing: B.Z., P.C., F.C., J.G.C., I.v.d.V, E.C., Y.C., Q.Z., and K.H.

## Competing interests

The authors declare no competing interests.
