## [Transparent Peer Review file · Nature Communications]

Enhanced CH₄ emissions from global wildfires likely due to undetected small fires

Corresponding Author: Dr Bo Zheng

Version 0:

Reviewer comments:

Reviewer #1

(Remarks to the Author)

Zhao et al have used an updated emission inventory for wildfires, including a small-fire correction, and used it as a prior for a Bayesian inversion of MOPITT satellite observations of carbon monoxide (CO). The focus of this study is to understand better the role of fires on recent changes in atmospheric methane. The authors make the link between CO and methane using emission factors inferred from field campaigns. They argue that their results point to a possible latitudinal redistribution of fire emissions of methane. I have provided my comments below. There are substantial gaps in the explanation of the work and/or methodology that need to be addressed.

There are a number of gas-phase pollutants that can act as tracers of incomplete combustion, including CO. The authors focus on the advantages of using CO. It has an atmospheric lifetime of weeks to months, against oxidation by the hydroxyl radical (OH), depending on latitude and season. While this makes it easier to observe elevated concentrations, compared with methane, interpreting CO data will rely more on understanding the OH chemistry. The authors will no doubt be aware that non-methane hydrocarbons and oxygenated compounds co-emitted with CO during combustion will also be oxidized rapidly to provide a secondary source of CO, sometimes contributing substantially to local and regional CO budgets. Without consideration of that additional source of CO in the wildfire emission inventories the discrepancy between the inventory and the satellite observation could be erroneously ascribed to primary emissions. After consulting with wildfire inventory experts, it is this reviewer's understanding that these inventories typically do NOT account for this secondary source of CO. Consequently, the authors need to explain how they overcome this issue in their model calculations. Currently, the manuscript is very scant on details about the atmospheric chemistry model being used or the inverse method. Given the information available, this reviewer thinks there is insufficient material to make the claim described in the study title.

The authors have translated changes in posterior CO emissions, inferred from MOPITT satellite data, to methane emissions using emission factors extracted from field campaign data. To their credit, the authors have done a valiant job but the campaign data available are sparse so this reviewer questions the substantial uncertainties associated with these emission factors. It is well known these emission factors can vary substantially throughout a burning season, which will not necessarily be captured by the available field campaign data. The authors make a number of arguments about incorrect fuel loading and consumption in inventories but again I think they need to consider the secondary source of CO from the oxidation of co-emitted non-methane hydrocarbons. This is particularly relevant for interpreting aircraft data that are sometimes collected far from the point of emission (e.g. ATom) and therefore subject to (sometimes) substantial chemical processing.

The reported trend in global fire methane emissions on line 124 is not statistically significant. I suspect the uncertainty associated with this value is higher than reported. Comparison of results from this study to Worden et al do include uncertainties – this needs to be rectified.

The discussion on line 250 about the new burned area product is interesting. They describe a 61% increase in burned area. Where has the burned area increased? Where it has increased will influence the fuel loading and subsequently the emission of CO. My understanding after talking to inventory experts is that this increase was the result of a bias correction based on a recalibration of the burned area that was then applied everywhere. I am not sure this necessarily supports the

main claims of the paper.

A subject for later, but shouldn't the availability of data not be contingent on contacting the authors? Surely, once data are published it becomes publicly available?

Reviewer #2

(Remarks to the Author)

NCOMMS-24-26464-T

Comments on: Zhao et al – Enhanced CH₄ emissions from global wildfires due to undetected small fires".....

General remarks

This is a major and globally important study, of considerable interest. It is certainly of Nature-level quality and significance. I recommend publication after minor revision.

Global wildfires are monitored by satellites and on the ground but small fires can be missed in Africa in particular, and also in wide areas in the high northern latitudes. This study uses CO as a means of quantifying the undetected smaller fires, and unsurprisingly discovers that they have a significant impact.

The methodology uses CO inversion, assuming both gases result from incomplete combustion of biomass.

The study is computer-based, using databases like the GFED (Global fire emissions database) and does not present original new measurements. I am not expert in this but the modelling analysis appears competent and the choice of data input and interpretation of findings seem reasonable. The work is well presented.

I have some qualitative concerns based on my own on-the-ground experience.

1. There seems to be little discussion of actual measurements of fire emissions, especially from the key locations in Africa. For example, the important aircraft work of Barker et al. over forest fires in southern Senegal is not mentioned.

Barker et al, (2020) Airborne measurements of fire emission factors for African biomass burning. Atmos. Chem. Phys., 20, 15443–15459 <https://doi.org/10.5194/acp-20-15443-2020>

2. There is no mention of the major impact on CO emissions of the introduction of catalytic converters in cars in the late 1990s-early 2000, and the simultaneous 'dieselisation' of many car fleets in Europe and elsewhere in the same decade. Both these factors had dramatic impacts on CO emissions. (e.g. see Zellweger et al. 2009. Inter-comparison of four different carbon monoxide measurement techniques and evaluation of the long-term carbon monoxide time series of Jungfraujoch. Atmospheric Chemistry and Physics, 9,3491-3503, and also Lowry et al. (2016) Marked long-term decline in ambient CO mixing ratio. Scientific Reports 6:25661 | DOI: 10.1038/srep25661). Lowry et al. were wrong in attributing the CO drop to catalytic clean up as the diesel emission scandal had not been widely publicised, but the CO drop was real.

I doubt there are many CO observations but Africa has a lot of cars!

3. In Africa there is anecdotal evidence for very widespread small crop waste fires and landfill fires (e.g. see Fig 14 in Nisbet, E.G., et al 2020. Methane mitigation: methods to reduce emissions, on the path to the Paris agreement. Reviews of Geophysics, 58(1), p.e2019RG000675). In India also, landfill fires are widespread. Moreover there is anecdotal evidence that the incidence of these crop waste and landfill fires has grown dramatically in the past two decades. While large-area seasonal tropical grass fires may have declined somewhat, the small crop waste fires and urban landfill smokes have probably more than tracked population growth – more people, more crops, and also there is less space for shifting cultivation so crop waste now has to be cleared..

Possibly the authors may find some space to comment on these points.

Detailed comments

Line 57 – for isotopes should specify an original source and mention the difference between C₃ (tree,bushes, lighter) and C₄ (grasses, heavy) vegetation sources. For example see the table 1 in MOYA/ZWAMPS Team, Philosophical Transactions of the Royal Society A 380, no. 2215 (2022): 20210112

Line 130-131 see also L273– annual mean biomass burning emissions? the manuscript in many places is not clear when the text is talking about fire emission and when it is mentioning total global or global anthropogenic emissions. In Line 273 the initial reading of the text seems to mean 78% of total anthropogenic emissions from ALL sources (including fossil fuels, cows and the rest!). It's confusing. Be specific.

Line 142 – decadal decrease. Are you sure that's not in part due to the drop in vehicle emissions? Africa has very large numbers of new cars!! Maybe the AIRS record would help.

Line 152 – worsening drought in the Arctic. That's complicated and not a fair generalisation. What is more relevant is the moisture balance, which is temperature dependent and also the seasonality of precipitation. The Arctic is warming. That's why it has more fires.

Line 164 – "Late fire season" this is very ambiguous. In outer tropical NH Africa the fire season is Nov-April, in SH outer tropical Africa it is July-Sept. In equatorial Africa with double dry seasons it depends on the latitude. In extratropical Africa it is Jan/Feb in the south and July/Aug in the north. So what does the text imply by 'late fire season'?? I'm lost!!

Line 178-180 – I think the Barker et al paper should be mentioned here (see point 1 above).

Line 170-181 – there seems to be no mention of landfill and heap fires. These are extremely widespread in Africa, especially around the new megacities – e.g. Lusaka has 3 million, Kinshasha has nearly 20 million - but also probably near every smallish village nowadays, and landfill fires are also common in India. Such fires, often only a few m² in area and without visible flames, are highly productive for CO and CH₄ but not readily visible in satellite databases like GFED.

Line 485 – 34,189 ppb HCN. Do you mean 34 ppm, as in English? Seems suicidally high! Or is this the 'continental comma'

– is it 34.189 ppb? in which case use the scientific decimal point. Incidentally see the Barker paper (cited above) on HCN

Overall Recommendation

The paper is important and of global significance. I recommend publication after minor revision.

Reviewer #3

(Remarks to the Author)

Review of: Enhanced CH₄ emissions from global wildfires due to undetected small fires (Zhao et al. 2024)

General comments:

The study by Zhao et al. shows an approach to estimate methane (CH₄) emissions from global wildfires between 2003 to 2020. The two-step methodology uses: (1) inverse modelling that assimilates satellite carbon monoxide (CO) observations from MOPITT, and (2) the observation-optimized CO emissions are then used to derive CH₄ emissions using CH₄/CO emissions factor ratios determined from literature field measurements. The field measurements (Supplementary Table 3 and Figure 3) have approximately an order of magnitude range between 0.009 to 0.085 but are stratified into the appropriate biome class (temperate forest, tropical forest, boreal forest, peatland and agricultural residues).

The results are interesting and relevant for the scientific community. The synthesis of satellite CO observations, established Bayesian inverse modelling, and field observations is excellent work and valuable for the scientific community. I can recommend the manuscript for publication with minor revisions.

1. I would suggest an alternative to referring to the CH₄ emissions derived in this work as “CO inversion-based estimation” throughout the figures and text. According to the authors’ description in the abstract, for the analysis they “reconstruct global fire CH₄ emissions by integrating satellite-observed carbon monoxide (CO)–based atmospheric inversion 30 with well-constrained fire CH₄ to CO emission ratio maps”. This is a two-step process that benefits from bridging results from both modelling (previously developed) and fieldwork (from literature). Referring to the results as “CO inversion-based” may cause some confusion as to whether these are results from a multi-species inversion of CO and CH₄ emissions through a general carbon simulation.

2. The results don’t appear to consider uncertainties in the posterior CO emissions from the inverse model, which are certainly nonzero. There isn’t as extensive detail regarding the CO inversion as one would expect from a modelling focused study, such as a description of an ensemble of model configurations and how that may represent the underlying uncertainties in the model. This is important to include because uncertainties in the posterior CO emissions should be propagated into uncertainties in the CH₄/CO emissions factor ratios. I can appreciate that this work has been done elsewhere and that extensive model testing is not the focus of this study. The authors state in the conclusions:

“This study is subject to potential uncertainties associated with multiple factors, mainly involving the global CO inversion and the fire CH₄/CO ERs developed based on field measurement data. The atmospheric CO inversion system benefits from the short atmospheric lifetime of CO and the reliability of satellite CO column retrieval. The CO inversion system was previously evaluated, demonstrating a substantial improvement in CO concentration simulation compared to independent CO observations. Although uncertainties remain, they may not change the main conclusions obtained herein.”

It is perhaps valid that uncertainties in the CO inversion would not change the qualitative conclusions of this study. However, the study is giving a quantitative conclusion on the CH₄ emissions, that they “tend to be underestimated by 27%”, so this estimate needs an associated uncertainty estimate to be robust. The approach of the authors towards this would imply that the inversion is not the central result, but rather the process of connecting a previously established model to literature fieldwork is the central result. The choices in naming convention for the results and overall communication should be clearer on this.

3. A central conclusion of this study is that the discrepancy between the CO inversion + CH₄/CO ratio CH₄ emissions and prior wildfire emissions are due to undetected small fires, as the title of the manuscript states. The authors justify this based on the high bias of the results compared to previous fire CH₄ estimates, and then use: (1) FIREX-AQ and ATom field campaign observations to show the CH₄/CO ratios are likely not overestimated and use (2): a comparison to high resolution FireCCISFD11 burned areas maps in sub-Saharan Africa to show that coarse resolution datasets are likely causing underestimations. I appreciate the well-reasoned argument here which can convince the reader the conclusion is likely to be true, but the evidence is not sufficient for the conclusion to be certain. The language in the manuscript should reflect this, and perhaps the title as well because the title reads with high certainty.

I am requesting minor revisions on these issues, in particular a) the presentation of uncertainties from the CO inversion and how this propagates into the uncertainty in the overall results and b) adjustments to the communication of scientific results in the manuscript.

Signed,
Sabour Baray

Please see the specific comments below.

Specific Comments:

L52: Do you mean 'main' source of pyrogenic CH₄ emissions

L61: The range (min – max estimates) is more useful to communicate than the percent difference

L80: "Atmospheric inversion" can be easily confused with the meteorological phenomena of a temperature inversion. It would be better to refer to this as "inverse modelling".

L82: High background CH₄ levels are ubiquitous in the atmosphere, so spatial overlap is redundant here. This entire sentence can be simplified since the three clauses are essentially referring to the same problem.

L97: This sentence is confusing because it sounds like the CH₄ emissions are computed in the modelling setup (i.e. a multi-species inversion), when in reality the CO emissions are being postprocessed with conversion factors.

L107: Again, it does not seem correct to refer to the results as "satellite-observed CO inversion-based" estimates, given how dependent the outcomes are on the second step of converting optimized CO emissions to CH₄ emissions.

L121: What does the comparison look like with FINN v2.5?

L175: There should be an attempt to quantify these potential uncertainties.

L206: It is difficult to conclude this confidently when you haven't tested the CH₄/CO ERs extensively over every biome.

L243: It is better to use "may be underestimating....due to...."

L256: Please see the general comments on this paragraph.

L273: This is a confusing metric, are you quantifying the difference between your result and the 4 models masked for each country, and then dividing by that country's total anthropogenic emissions?

L437: The full name of MOPITT should be given at least once.

L515: How long was the transport time?

Figure 1a: The shaded region indicating the uncertainties in CH₄/CO emission ratios appears to be a constant relative error, is this the case? This needs to be explained more clearly.

Figure 2a: I find the gray dashed differences line to be distracting and redundant with the red and green curves already present.

Supplementary Table 1: The CO inversion-based estimates should have an error interval associated as they are presented in Supplementary Table 3 for the EFs.

Supplementary Figure 11: The caption for this figure is wordy and difficult to understand. It may help with interpretation to have the anthropogenic CH₄ emissions as well as the fire CH₄ emissions on top of the bars, followed by the percentages to follow the calculations better.

Version 1:

Reviewer comments:

Reviewer #1

(Remarks to the Author)

Reviewer #2

(Remarks to the Author)

This is an interesting and potentially important paper. The authors have responded well to my comments, though I still have some minor quibbles (see below). I also remain concerned that the authors' image of moist tropical Africa as a quasi-natural land surface environment, not the reality of widespread intense farming, many huge new cities, and abundant cars, trucks and small smouldering fires. That said, the paper should now be accepted and should go forwards to publication.

Minor points:

Line 58 on says "Moreover, CH₄ emissions generated by fires are isotopically heavier than those of biogenic and thermogenic origin, with C₃ vegetation (e.g., trees) producing lighter isotopes and C₄ vegetation (e.g., grasses) producing heavier ones."

Two comments

1. Delete 'and thermogenic' – 'thermogenic' means 'fire-generated' (although here used in the sense of geological emissions. Moreover many geological (thermogenic) methane emissions are isotopically heavy, and some geological emissions (e.g. from deep coal mines) can even be heavier than fire emissions (particularly those from burning C₃ plants)

2. Poor language – 'heavier ones' What does that mean?

Maybe change these lines to something like

"Moreover, CH₄ emissions generated by fires are isotopically heavier than methane of biogenic origin, as methane from burning C₃ vegetation (e.g. trees), and especially from burning C₄ vegetation (e.g. many tropical grasses, maize, sugar cane), is typically enriched in ¹³C compared to biogenic emissions from wetlands, ruminants, or waste.

Line 228 has a spelling error 'emissioin'.

Reviewer #3

(Remarks to the Author)

Thank you to the authors for providing revisions on the manuscript "Enhanced CH₄ emissions from global wildfires likely due to undetected small fires", including a revision of the title to better represent the study conclusions. My concerns on 1) the inclusion of an uncertainty estimate for the inverse-modelling results, and 2) the scientific communication with respect to the methodology, results, and conclusions of this modelling study have been addressed. I am happy to recommend this manuscript for publication.

Responses to reviewers' comments

We appreciate the reviewer's careful and thoughtful comments on our manuscript entitled "*Enhanced CH₄ emissions from global wildfires due to undetected small fires*" and thank the helpful suggestions to improve the article. We have carefully reviewed all comments and revised the article accordingly. The sentences are depicted in red in the manuscript text to highlight the new addition and used strikethrough for deletion. All the responses are in the green background below.

Responses to Reviewer 1 comments:

1. Zhao et al have used an updated emission inventory for wildfires, including a small-fire correction, and used it as a prior for a Bayesian inversion of MOPITT satellite observations of carbon monoxide (CO). The focus of this study is to understand better the role of fires on recent changes in atmospheric methane. The authors make the link between CO and methane using emission factors inferred from field campaigns. They argue that their results point to a possible latitudinal redistribution of fire emissions of methane. I have provided my comments below. There are substantial gaps in the explanation of the work and/or methodology that need to be addressed.

Response:

Thank you so much for all your insightful comments and suggestions. We understand your concerns and have done our utmost to address and clarify the issues you raised. Please find our detailed responses below.

2. There are a number of gas-phase pollutants that can act as tracers of incomplete combustion, including CO. The authors focus on the advantages of using CO. It has an atmospheric lifetime of weeks to months, against oxidation by the hydroxyl radical (OH), depending on latitude and season. While this makes it easier to observe elevated concentrations, compared with methane, interpreting CO data will rely more on understanding the OH chemistry. The authors will no doubt be aware that non-methane hydrocarbons and oxygenated compounds co-emitted with CO during combustion will also be oxidized rapidly to provide a secondary source of CO, sometimes contributing substantially to local and regional CO budgets. Without consideration of that additional source of CO in the wildfire emission inventories the discrepancy between the inventory and the satellite observation could be erroneously ascribed to primary emissions. After consulting with wildfire inventory experts, it is this reviewer's understanding that these inventories typically do NOT account for this secondary source of CO. Consequently, the authors need to explain how they overcome this issue in their model calculations. Currently, the

manuscript is very scant on details about the atmospheric chemistry model being used or the inverse method. Given the information available, this reviewer thinks there is insufficient material to make the claim described in the study title.

Response:

Thank you for your valuable suggestions. As you pointed out, attributing changes in the tropospheric CO burden requires an accurate representation of both sources and sinks of OH. Additionally, about half of the tropospheric CO burden results from the chemical oxidation of CH₄ and volatile organic compounds (VOCs)^{1,2} such as formaldehyde, acetaldehyde, glyoxal, and methylglyoxal. Indeed, the bottom-up fire emission inventory used for our inversion modeling does not account for this secondary source of CO. Moreover, the other anthropogenic emission inventories employed in this study also lack this secondary CO source. However, it's important to note that the atmospheric CO concentrations simulated by the atmospheric chemistry model for each grid box reflect the sum of multiple CO emission sources minus the CO sink. This implies that the simulated atmospheric CO concentrations, which are calibrated against satellite observations, inherently include the impacts of both direct CO emissions from various sectors and the secondary CO production from the oxidation of CH₄ and VOCs. We acknowledge that our initial manuscript may have provided an overly concise explanation of the CO emission inversion methodology, which could have led to potential confusion among readers and reviewers. To address this, we have added the following statement to **Methods** section: "*The methodology for CO inversion is detailed in Supplementary Text 1.*". Supplementary Text 1 has been included in the Supplementary Information as follow:

"Supplementary Text 1: Methodology of atmospheric inversion

We employ the global 3-D transport model from Laboratoire de Météorologie Dynamique (LMDz), coupled with the Simplified Atmospheric Chemistry Assimilation System (SACS)³, to simulate atmospheric CO dynamics in a 3-D grid which can be described by equation (1):

$$\begin{aligned}
\frac{\partial[\text{CO}]}{\partial t} &= \sum (\text{Source}_{\text{CO}}) - \text{Sink}_{\text{CO}} \\
&= -v \cdot \nabla[\text{CO}] + \sum_{\text{sector}} (E_{\text{CO}}) + P_{\text{CH}_4 \rightarrow \text{CO}} + P_{\text{NMVOCs} \rightarrow \text{CO}} - k_{\text{CO}+\text{OH}}(T)[\text{CO}][\text{OH}] - \text{Dep}_{\text{CO}}
\end{aligned} \tag{1}$$

The temporal evolution of CO concentrations ($\partial[\text{CO}]/\partial t$) in each grid cell is represented as the balance of CO emissions ($\text{Source}_{\text{CO}}$) and CO removal (Sink_{CO}). Where, the flux divergence term ($v \cdot \nabla[\text{CO}]$) accounts for CO transport across grid cells, which sums to zero globally. Surface emissions (E_{CO}) originate from sources like anthropogenic, biogenic, biomass burning and oceanic sectors. Chemical production of CO from CH_4 and non-methane volatile organic compounds (NMVOCs), driven by OH oxidation ($P_{\text{CH}_4 \rightarrow \text{CO}}$ and $P_{\text{NMVOCs} \rightarrow \text{CO}}$), is included, alongside CO's chemical sink ($k_{\text{CO}+\text{OH}}(T)[\text{CO}][\text{OH}]$), which depends on temperature (T), CO concentration ($[\text{CO}]$), and OH concentration ($[\text{OH}]$). Dry deposition (Dep_{CO}) contributes approximately 7% to the total CO sink².

The atmospheric Bayesian inversion framework is built upon the LMDz-SACS model^{4,5}, and satellite observations of relevant gases are assimilated to constrain the system⁶⁻⁸. The inversion inference is formulated as a variational optimization, minimizing the following cost function:

$$J(x) = (x - x^b)^T B^{-1} (x - x^b) + (H(x) - y)^T R^{-1} (H(x) - y) \tag{2}$$

The control vector, x , contains the target variables, with x^b as the prior estimate, assuming Gaussian error statistics with covariance B . The observation vector, y , holds assimilated data, and its errors are also assumed Gaussian with covariance R . H is the forward model (a combination of the LMDz-SACS model, sampling operator, and averaging kernel) computes the observation equivalent from x . R incorporates measurement, forward model, and representation errors. The inversion is solved iteratively using forward and adjoint codes until cost function convergence, yielding an optimized model state that fits all constraints within their uncertainties. This framework comprehensively models the hydrocarbon oxidation chain, including primary CO emissions, secondary CO production, and chemical sinks, with tracers CH_4 , formaldehyde (HCHO), CO, and intermediate species,

with chain reactions driven by OH among all chemical species. Methyl chloroform (MCF) is also included to constrain OH concentrations. Variable configurations are detailed in previous studies⁶⁻⁸."

3. The authors have translated changes in posterior CO emissions, inferred from MOPITT satellite data, to methane emissions using emission factors extracted from field campaign data. To their credit, the authors have done a valiant job but the campaign data available are sparse so this reviewer questions the substantial uncertainties associated with these emission factors. It is well known these emission factors can vary substantially throughout a burning season, which will not necessarily be captured by the available field campaign data. The authors make a number of arguments about incorrect fuel loading and consumption in inventories but again I think they need to consider the secondary source of CO from the oxidation of co-emitted non-methane hydrocarbons. This is particularly relevant for interpreting aircraft data that are sometimes collected far from the point of emission (e.g. ATom) and therefore subject to (sometimes) substantial chemical processing.

Response:

We agree with your comments. As noted in the *Discussion and Implication* section, the sparsity of observational data used to validate the CH₄/CO ratios in this study may limit our ability to fully capture the variations in emission factors across different fire phases and combustion efficiencies over the course of a burning season, which is indeed a limitation of our study. To address this, we quantified the impact of uncertainties in the 148 CH₄/CO ratios we collected on emission calculations, as described in the *Methods* section: "*To assess the uncertainties associated with CO inversion-based CH₄ emissions (shaded area in Fig. 1), we substituted the average CH₄/CO emission factor ratio with the standard deviation of the CH₄/CO emission factor ratios for conversion factors (CFs) of boreal forest and peatland, and replaced the regression slope with the standard deviation of the residuals for other fire types.*". Additionally, in response to the third reviewer's suggestion, we have also incorporated the uncertainties associated with the CO inversion. The results of these uncertainties assessment are now reflected in the revised Fig. 1(a), represented by the gray shaded area.

Notably, even the lower boundary of the uncertainty range (the bottom edge of the gray shaded area) yields a CH₄ emission estimate of 18.1 Tg yr⁻¹ (average for 2003–2020), which closely aligns with the average estimates from four global fire emission models (18.9 Tg yr⁻¹ for the same period). Moreover, in most years, the lower boundary also exceeds the average of the four models' estimates.

Thus, while we acknowledge the uncertainties arising from CH₄/CO ratios and recognize the challenges in validating them due to sparse observational data, our uncertainty analysis indicates that these uncertainties are unlikely to substantially alter the main conclusions of this study. To clarify this point, we revised relevant paragraph (Line 285–296) in the **Discussion and Implication** section as follow: "*The atmospheric CO inversion system benefits from the short atmospheric lifetime of CO and the reliability of satellite CO column retrieval. The CO inversion system was previously evaluated, demonstrating a substantial improvement in CO concentration simulation compared to independent CO observations. Regarding CH₄/CO ERs, evaluation against aircraft measurements revealed a close agreement between field-measured values and the data employed herein. Although uncertainties persist, stemming from the inversion process, limited spatiotemporal coverage of the evaluation datasets (e.g., the FIREX-AQ and ATom campaigns were only conducted in summer and winter, respectively) and lack of peat fire plume observations. Despite these uncertainties, the lower bound of the uncertainty range (averaging 18.1 Tg yr⁻¹ for 2003–2020) remains in close alignment with the average estimate from four global fire emission models (18.9 Tg yr⁻¹ for the same period), suggesting that they are unlikely to significantly alter the study's main conclusions of this study*".

Regarding the secondary source of CO, please refer to our response to the second comment. Additionally, when validating the EF ratio based on ATom observation data, we acknowledge that the distance between the observation points and emission sources introduces non-negligible chemical losses during transport. Therefore, this study has applied a first-order correction for these losses, following established methods from previous study⁹. The details of the calculation are provided in the **Method** section as follows: "*Since the plume transport*

time was relatively long, we calculated first-order losses of CH₄ and CO and converted the concentration enhancement ratio of CH₄ to CO to the CH₄/CO ER, as described by Lutsch et al.⁹. This conversion process can be represented by the following equation...".

4. The reported trend in global fire methane emissions on line 124 is not statistically significant. I suspect the uncertainty associated with this value is higher than reported. Comparison of results from this study to Worden et al do include uncertainties – this needs to be rectified.

Response:

Yes, our estimated trend of fire CH₄ emissions from 2003 to 2020 ($-1.1\% \pm 1.2\% \text{ yr}^{-1}$) is not statistically significant, primarily due to the abrupt increase in emissions observed in 2015. However, we would like to clarify that the uncertainty associated with this trend is represented by the 95% confidence interval from the nonparametric Mann–Kendall test. After thoroughly reviewing the article by Worden et al.¹⁰, we noted that they did not calculate the uncertainty for their reported trend. However, in the first section of the **Results**, they mention the following regarding uncertainty: “...we find that mean 2001–2014 emissions average to $12.9 \pm 3.3 \text{ Tg CH}_4$ per year, and the decrease averages to $3.7 \pm 1.4 \text{ Tg CH}_4$ per year for 2008–2014, relative to 2001–2007.”

Although Worden et al.¹⁰ do not provide a clear explanation of how the ± 3.3 and ± 1.4 Tg values were derived, we computed these figures based on the pink-shaded uncertainty region (CH₄/CO uncertainty) in their Fig. 1(as shown below) and found a close match with their reported numbers. To ensure a consistent comparison with the uncertainty range described in their main conclusion (“the decrease averages to $3.7 \pm 1.4 \text{ Tg CH}_4$ per year for 2008–2014, relative to 2001–2007”) and considering this difference is primarily attributed to changes occurring between 2006 and 2008, we applied a similar approach using the grey-shaded uncertainty region in our Fig. 1(a) (uncertainties in CH₄/CO emission ratios). Based on this calculation, we have revised the corresponding paragraph (Line 139-141) as follow: “During a similar period, our study estimated a decrease in the annual mean CH₄ emissions

of $4.3 \pm 1.0 \text{ Tg CH}_4 \text{ yr}^{-1}$ from 2003–2007 to 2008–2012, which was similar to that estimated by Worden et al.²¹ and 64% higher than that of the GFED v4.1s model. "

[Figure redacted]

Fig. 1 Trend of methane emissions from biomass burning. Expected methane emissions from fires based on the Global Fire Emissions Database (black) and the CO emissions plus CH₄/CO ratios shown here (red). The range of uncertainties in blue is due to the calculated errors from the CO emissions estimate and the shaded red describes the range of error from uncertainties in the CH₄/CO emission factors. (source from Worden et al.¹⁰)

5. The discussion on line 250 about the new burned area product is interesting. They describe a 61% increase in burned area. Where has the burned area increased? Where it has increased will influence the fuel loading and subsequently the emission of CO. My understanding after talking to inventory experts is that this increase was the result of a bias correction based on a recalibration of the burned area that was then applied everywhere. I am not sure this necessarily supports the main claims of the paper.

Response:

The new GFED5 burned area product is primarily based on finer-resolution satellite imagery from Landsat and Sentinel-2, combined with land use data for calibration. The main improvement after calibration lies in GFED5's ability to capture a broader range of small-scale fires. To evaluate its quality, GFED5 has been compared with regional burned area products derived from fine-resolution satellite imagery (as shown in Fig. 11 of Chen et al. (GFED v5)¹¹). These comparisons reveal a significant reduction in bias and notable improvements in correlation. The African burned area product used in this study for comparison with inversion results is the same as the one used in these assessments. Moreover, Chen et al. (GFED v5)¹¹

mentions that "the burned area data in several widely used global emission inventories are substantially underestimated, largely due to the difficulty of detecting and measuring burned area associated with small fires," and further states that "the inclusion of small-fire contributions in the GFED5 burned area product is likely to reduce the seasonal variability of fire emissions, making it more consistent with atmospheric-based estimates." These findings align with our conclusion that existing fire emission models relying on coarse-resolution burned area data may underestimate emissions due to their failure to account for small fires.

We recognize that our initial explanation may have led to confusion among readers and reviewers, and have thus revised the text as follows: "It should be noted that a recently developed burned area product by Chen et al. (GFED v5) reported an approximate 61% increase in global burned area compared to GFED v4.1s, after adjusting for commission and omission errors, particularly those related to small fires. This indicates that fire emissions based on GFED v4.1s were substantially underestimated, supporting the findings presented in this study."

Figure 11. A spatial ($0.25^\circ \times 0.25^\circ$) comparison of burned area from GFED5 with burned area from MCD64A1 and higher-resolution satellites (SFD for Africa, MAPB for Brazil, IDNS2 for Indonesia; see Table 1 for details) is shown for (a) Africa, (b) Brazil, and

(c) Indonesia. (source from Chen et al.¹¹)

6. A subject for later, but shouldn't the availability of data not be contingent on contacting the authors? Surely, once data are published it becomes publicly available?

Response:

If our responses to the reviewers' comments are deemed satisfactory by both the reviewers and the editor, and the paper is accepted for publication, we will make the reconstructed global fire CH₄ emission data publicly available through the Zendo database link. We believe this will enhance the scientific community's understanding of the global CH₄ cycle and its broader implications.

Responses to Reviewer 2 comments:

1. This is a major and globally important study, of considerable interest. It is certainly of Nature-level quality and significance. I recommend publication after minor revision. Global wildfires are monitored by satellites and on the ground but small fires can be missed in Africa in particular, and also in wide areas in the high northern latitudes. This study uses CO as a means of quantifying the undetected smaller fires, and unsurprisingly discovers that they have a significant impact. The methodology uses CO inversion, assuming both gases result from incomplete combustion of biomass. The study is computer-based, using databases like the GFED (Global fire emissions database) and does not present original new measurements. I am not expert in this but the modelling analysis appears competent and the choice of data input and interpretation of findings seem reasonable. The work is well presented.

Response:

We really appreciate all your encouraging comments and constructive comments. We have carefully addressed and revised the manuscript based on the points you raised. We hope these revisions meet with your approval and further strengthen the quality of the study.

2. There seems to be little discussion of actual measurements of fire emissions, especially from the key locations in Africa. For example, the important aircraft work of Barker et al. over forest fires in southern Senegal is not mentioned. Barker et al, (2020) Airborne measurements of fire emission factors for African biomass burning. *Atmos. Chem. Phys.*, 20, 15443–15459 <https://doi.org/10.5194/acp-20-15443-2020>

Response:

As suggested, we have thoroughly reviewed the article by Barker et al.¹² and identified key data regarding aircraft measurements of near-field biomass burning plumes over the Casamance region in the wooded savannah of southwest Senegal. The CH₄/CO emission ratio (ER) reported in their study is 0.04–0.05 ppb/ppb, consistent with the ER ratio of 0.04 ppb/ppb used for the savanna biome in our study. In response, we have added the following statement

to the third paragraph (Line 215) of the section *Undetected small fires explain higher CO inversion-based CH₄ emissions*: "Additionally, aircraft measurements³⁸ of near-field fire plumes over the savanna region in Senegal reported CH₄/CO ER of 0.04–0.05 ppb/ppb, consistent with the 0.04 ppb/ppb ER used for the savanna biome in this study. "

3. There is no mention of the major impact on CO emissions of the introduction of catalytic converters in cars in the late 1990s–early 2000, and the simultaneous ‘dieselisation’ of many car fleets in Europe and elsewhere in the same decade. Both these factors had dramatic impacts on CO emissions. (e.g. see Zellweger et al. 2009. Inter-comparison of four different carbon monoxide measurement techniques and evaluation of the long-term carbon monoxide time series of Jungfraujoch. Atmospheric Chemistry and Physics, 9,3491–3503, and also Lowry et al. (2016) Marked long-term decline in ambient CO mixing ratio. Scientific Reports 6:25661 | DOI: 10.1038/srep25661). Lowry et al. were wrong in attributing the CO drop to catalytic clean up as the diesel emission scandal had not been widely publicised, but the CO drop was real. I doubt there are many CO observations but Africa has a lot of cars!

Response:

With the additional information and references you provided; we have a deeper understanding of CO emissions in the African region. Given the advancements in combustion technologies^{13,14} (e.g., the widespread adoption of catalytic converters) and the use of diesel¹⁵ in the early 21st century, global anthropogenic CO emissions have generally shown a decreasing trend over the past two decades¹⁶. However, our review of previous research indicates a slight upward trend in satellite-observed CO columns in Africa (Fig. 1 in Zheng et al.⁸). We found that emissions from concentrated anthropogenic sources are indeed increasing (Fig. 8 in Zheng et al. ⁸), a result consistent with other bottom-up inventories (CEDS,EDGAR) findings^{16,17}, where the growing residential sector is primarily responsible for the rising emissions in equatorial Africa. Conversely, our inversions revealed a decline in CO emissions from wildfire-affected areas(Fig. 6 in Zheng et al. ⁸), aligning with other bottom-up estimates and satellite observations of decreasing burned area¹⁸, mainly attributed to reductions in

grassland burning driven by human-induced land use changes. We believe the upward trend in anthropogenic emissions will not affect the declining trend in wildfire emissions, as both are supported by independent data evidence.

As you mentioned, the prevalence of diesel-powered vehicles in Africa, which emit little CO but are rich in NO_x¹⁵, could indeed be a major factor driving the decline in road transport CO emissions. However, the increase in NO_x emissions may exacerbate photochemical pollution, presenting an important scientific issue worth further exploration. While this is an important topic, it differs somewhat from the primary focus of our current study. We would prefer to address this issue in future research.

4. In Africa there is anecdotal evidence for very widespread small crop waste fires and landfill fires (e.g. see Fig 14 in Nisbet, E.G., et al 2020. Methane mitigation: methods to reduce emissions, on the path to the Paris agreement. *Reviews of Geophysics*, 58(1), p.e2019RG000675). In India also, landfill fires are widespread. Moreover, there is anecdotal evidence that the incidence of these crop waste and landfill fires has grown dramatically in the past two decades. While large-area seasonal tropical grass fires may have declined somewhat, the small crop waste fires and urban landfill smokes have probably more than tracked population growth – more people, more crops, and also there is less space for shifting cultivation so crop waste now has to be cleared.

Response:

Thank you for your feedback. As you noted and based on findings from Nisbet et al.,¹⁹ small-fire types such as landfill burning and crop residue fires are often inadequately managed in many developing countries, resulting in increased methane emissions and the release of other harmful pollutants. Technically, satellite burned-area-based wildfire emission products, which include agricultural sources²⁰, but may not fully capture the complexities associated with small fires²¹, leading to potential omissions. However, from a top-down inversion perspective, emissions from these small fires are likely captured through the CO plumes detected by satellite trace gas measurements, which we consider a strength of our study.

We have added the following paragraphs (Lines 272 and 309) to the **Discussion and Implication** section to expand on the small-fire topic: "*Emissions from small-fire types (e.g., landfill and crop residue burning) may not be accurately captured by burned-area-based products²¹; however, their plumes are likely included in satellite CO observations, which are used to constrain fire emissions in our inversion system.*" and "*Due to inadequate management of emissions from small-fire types (e.g., landfill and crop residue burning) in developing countries, which are exhibiting obvious growth trends in certain regions²², leads to increased CH₄ emissions and the release of other harmful gases and particulates. Without improved regulation, future may further exacerbate such emissions¹⁹.*"

5. Line 57 – for isotopes should specify an original source and mention the difference between C3 (tree, bushes, lighter) and C4 (grasses, heavy) vegetation sources. For example see the table 1 in MOYA/ZWAMPS Team, Philosophical Transactions of the Royal Society A 380, no. 2215 (2022): 20210112

Response:

As you recommended, we have revised the sentence as follow: "*CH₄ emissions generated by fires are isotopically heavier than those of biogenic and thermogenic origin²³, with C3 vegetation (e.g., trees) producing lighter isotopes and C4 vegetation (e.g., grasses) producing heavier ones²⁴.*"

6. Line 130-131 see also L273– annual mean biomass burning emissions? the manuscript in many places is not clear when the text is talking about fire emission and when it is mentioning total global or global anthropogenic emissions. In Line 273 the initial reading of the text seems to mean 78% of total anthropogenic emissions from ALL sources (including fossil fuels, cows and the rest!). It's confusing. Be specific.

Response:

Thank you for pointing out the need for clarification. We have thoroughly reviewed and revised the following sections for improved clarity:

*"Line 54 – ...4% of global **total** CH₄ emissions (**biogenic, thermogenic, and pyrogenic sources**) per year.*

*Line 139 – ...annual mean **fire** CH₄ emissions...*

Line 143 – ...Our CO-based fire CH₄ emissions...

Line 145 – The higher fire CH₄ emissions..."

In the original manuscript, Line 273 refers to "...ranging from 8% to 78%, of the total anthropogenic CH₄ emissions from the top 10 emitting countries..." which denotes the sum of anthropogenic CH₄ emissions from all sectors for each country as reported in the EDGAR database. To clarify, we have revised corresponding sentence (Line 304) as follow :
".....ranging from 8% to 78%, of the total anthropogenic CH₄ emissions (all sectors in the EDGARv7.0²⁵ database) from the top 10 emitting countries..."

7. Line 142 – decadal decrease. Are you sure that's not in part due to the drop in vehicle emissions? Africa has very large numbers of new cars!! Maybe the AIRS record would help.

Response:

In Line 142 of the original manuscript, we stated: "*The decadal decrease in fire CH₄ emissions from 2003–2011 to 2012–2020 revealed by our CO-based results was attributed to the pronounced decrease in fire emissions over the 30°S–15°N latitude band (Supplementary Fig. 2), in which satellites detected a decline in grassland burning due to population growth and agriculture expansion¹⁸.*". The "decadal decrease" refers specifically to the reduction in CH₄ emissions from fire sources. These fire CH₄ emissions were derived from inverted fire CO emissions, which, in turn, were isolated from the inversion estimates of total surface CO fluxes.

We recognize that during the inversion process, changes in emissions from various sectors (including vehicle emissions) may influence the inversion results for total CO fluxes, and thus, the results for different sectors could vary by region and time period. However, addressing these sectoral variations would likely require extensive sensitivity experiments,

which falls outside the primary scope of this study. As indicated in Supplementary Fig. 2, the most substantial decline in emissions occurred within the 30°S–15°N latitude band, aligning with previous studies that report reductions in grassland burning due to land use changes driven by human activity¹⁸. Additionally, as noted in our response to your third comment, we acknowledge that this is an important topic but one that differs from the main focus of our current study. We plan to address this issue in future research. We have also revised the corresponding sentence as follow: "*...in which satellites detected a decline in grassland burning due to population growth and agriculture expansion¹⁸, which are likely the main contributing factors.*".

8. Line 152 – worsening drought in the Arctic. That’s complicated and not a fair generalisation. What is more relevant is the moisture balance, which is temperature dependent and also the seasonality of precipitation. The Arctic is warming. That's why it has more fires.

Response:

We agree that attributing the increase in boreal fire emissions solely to worsening drought conditions oversimplifies the situation. We have revised the sentence in Line 161 as follows: "*However, boreal fire emissions increased since 2003 likely driven by changes in the moisture balance as the Arctic continues to warm, ...*".

9. Line 164 – “Late fire season” this is very ambiguous. In outer tropical NH Africa the fire season is Nov-April, in SH outer tropical Africa it is July-Sept. In equatorial Africa with double dry seasons it depends on the latitude. In extratropical Africa it is Jan/Feb in the south and July/Aug in the north. So what does the text imply by ‘late fire season’?? I’m lost!!

Response:

Thank you for pointing out the need for clarification. Based on our previous research⁷, we identified that the peak fire season, characterized by the maximum burned area, typically occurs in December and January in Northern Africa, and from July to September in Southern Africa. After these months, the burned area declines, marking the late fire season. Therefore,

the "late fire season" in our context refers to February, March, and November at 0°–15°N, and October at 30°S–0°. To clarify this, we have revised the corresponding sentence (Line 172) as follows: "*In addition, our atmospheric CO-based results indicate a larger allocation of fire CH₄ emissions to the late fire season (February, March, and November at 0°–15°N, and October at 30°S–0°).*"

10. Line 178-180 – I think the Barker et al paper should be mentioned here (see point 1 above).

Response:

We have carefully reviewed the article by Barker et al.¹² and included a comparative analysis of the CH₄/CO emission ratios measured by Barker et al.¹² with those used in our study. Please refer to our response to your second comment for detailed information.

11. Line 170-181 – there seems to be no mention of landfill and heap fires. These are extremely widespread in Africa, especially around the new megacities – e.g. Lusaka has 3 million, Kinshasha has nearly 20 million - but also probably near every smallish village nowadays, and landfill fires are also common in India. Such fires, often only a few m² in area and without visible flames, are highly productive for CO and CH₄ but not readily visible in satellite databases like GFED.

Response:

This suggestion is closely related to the fourth comment you provided. We have integrated the relevant discussion into the *Discussion and Implications* section. Please refer to our response to the fourth comment for more details.

12. Line 485 – 34,189 ppb HCN. Do you mean 34 ppm, as in English? Seems suicidally high! Or is this the 'continental comma' – is it 34.189 ppb? in which case use the scientific decimal point. Incidentally see the Barker paper (cited above) on HCN

Response:

Thank you for pointing out this issue. Upon reviewing the original data and associated documentation, we identified an error in the unit, which should be ppt instead of ppb for both

FIREX-AQ and ATom measurement data. We also conducted a thorough review of the units used for all other observational data in our study, confirming that only HCN was incorrectly labeled. The unit has now been corrected to ppt throughout the manuscript and supplementary information.

Responses to Reviewer 3 comments:

1. The study by Zhao et al. shows an approach to estimate methane (CH₄) emissions from global wildfires between 2003 to 2020. The two-step methodology uses: (1) inverse modelling that assimilates satellite carbon monoxide (CO) observations from MOPITT, and (2) the observation-optimized CO emissions are then used to derive CH₄ emissions using CH₄/CO emissions factor ratios determined from literature field measurements. The field measurements (Supplementary Table 3 and Figure 3) have approximately an order of magnitude range between 0.009 to 0.085 but are stratified into the appropriate biome class (temperate forest, tropical forest, boreal forest, peatland and agricultural residues). The results are interesting and relevant for the scientific community. The synthesis of satellite CO observations, established Bayesian inverse modelling, and field observations is excellent work and valuable for the scientific community. I can recommend the manuscript for publication with minor revisions.

Response:

Thank you very much for all your valuable comments and suggestions. We have carefully addressed each of the points you raised and made corresponding revisions to the manuscript. We hope these changes meet with your approval and further enhance the quality of the study.

2. I would suggest an alternative to referring to the CH₄ emissions derived in this work as “CO inversion-based estimation” throughout the figures and text. According to the authors’ description in the abstract, for the analysis they “reconstruct global fire CH₄ emissions by integrating satellite-observed carbon monoxide (CO)-based atmospheric inversion 30 with well-constrained fire CH₄ to CO emission ratio maps”. This is a two-step process that benefits from bridging results from both modelling (previously developed) and fieldwork (from literature). Referring to the results as “CO inversion-based” may cause some confusion as to whether these are results from a multi-species inversion of CO and CH₄ emissions through a general carbon simulation.

Response:

We agree with your concern regarding the potential confusion caused by the term "CO inversion-based." To address this, we have revised all relevant descriptions throughout the manuscript and Supplementary Information (text and figure), replacing "CO inversion-based" with "CO-based." For example, "*Our CO inversion-based fire CH₄ emissions*" has been changed to "*Our CO-based fire CH₄ emissions*."

3. The results don't appear to consider uncertainties in the posterior CO emissions from the inverse model, which are certainly nonzero. There isn't as extensive detail regarding the CO inversion as one would expect from a modelling focused study, such as a description of an ensemble of model configurations and how that may represent the underlying uncertainties in the model. This is important to include because uncertainties in the posterior CO emissions should be propagated into uncertainties in the CH₄/CO emissions factor ratios. I can appreciate that this work has been done elsewhere and that extensive model testing is not the focus of this study. The authors state in the conclusions:

"This study is subject to potential uncertainties associated with multiple factors, mainly involving the global CO inversion and the fire CH₄/CO ERs developed based on field measurement data. The atmospheric CO inversion system benefits from the short atmospheric lifetime of CO and the reliability of satellite CO column retrieval. The CO inversion system was previously evaluated, demonstrating a substantial improvement in CO concentration simulation compared to independent CO observations. Although uncertainties remain, they may not change the main conclusions obtained herein."

It is perhaps valid that uncertainties in the CO inversion would not change the qualitative conclusions of this study. However, the study is giving a quantitative conclusion on the CH₄ emissions, that they "tend to be underestimated by 27%", so this estimate needs an associated uncertainty estimate to be robust. The approach of the authors towards this would imply that the inversion is not the central result, but rather the process of connecting a previously established model to literature fieldwork is the central result. The choices in naming convention for the results and overall communication should be clearer on this.

Response:

Thank you for your valuable suggestions; we fully agree with your concerns. Given the difficulty in accurately estimating uncertainties for inversion products and the computational constraints that limit detailed consideration, Worden et al.¹⁰ also relied on empirical uncertainty estimates for top-down CO emissions, citing similar limitations. Here, we calculated the approximate uncertainty of inversion-based CO emissions using results from three sensitivity CO inversion simulations (2010–2017) from our previous study⁸. We then combined these estimates with the uncertainties in CH₄/CO ER ratios to derive an integrated uncertainty range. The *Method* section has been revised to reflect this approach, and Fig. 1 has been updated as below:

"We assess the uncertainties associated with CO-based CH₄ emissions (shaded area in Fig. 1), according to the equations (3):

$$\Delta E_{i,j,t}^{CH_4} = E_{i,j,t}^{CH_4} \times \sqrt{\left(\frac{\Delta E_{i,j,t}^{CO}}{E_{i,j,t}^{CO}}\right)^2 + \left(\frac{\Delta ER_{i,j,t}^{CH_4:CO}}{ER_{i,j,t}^{CH_4:CO}}\right)^2} \quad (3)$$

Where, ΔE and ΔER represent the uncertainties in emissions and emission ratios, respectively. The ΔER was derived based on equation (1), where we substituted the average CH₄/CO emission factor ratio with the standard deviation of the CH₄/CO emission factor ratios for CFs of boreal forest and peatland, and we replaced the regression slope with the standard deviation of the residuals for other fire types. The ΔE for inversion-based CO emissions was obtained by calculating the standard deviation of monthly and grid-based CO inversion results from three sensitivity simulations in our previous study (see Table 2 in Zheng et al.⁸). Subsequently, we evaluated the uncertainty range of ΔE and ΔER , and propagated such uncertainties to estimate the emission uncertainties."

Figure 1. Comparison between global CO-based fire CH₄ emission estimates and global fire emission model results. (a) Annual trends in fire CH₄ emissions from CO-based estimates (red curve) and average estimates of 4 global fire emission models (green curve) from 2003 to 2020, including the fitted linear trends (dashed red and green lines). The shaded gray region represents the range of error derived from uncertainties in CH₄/CO emission ratios and inversion-based CO estimates, which vary with changes in dry matter and CO estimates (Methods). Trend assessments are conducted using the nonparametric Mann–Kendall test and Theil–Sen estimator, with 2003–2020 trends and uncertainties provided. Significant trends are denoted by asterisks (* $p < 0.1$ and ** $p < 0.05$). (b) Spatial distribution of differences between CO-based CH₄ emission estimates and those of 4 fire emission models. Data averaged between 2003 and 2020 are at the spatial resolution of 3.75° longitude \times 1.9° latitude.

We have also revised relevant paragraph (Line 285–296) in the **Discussion and Implication** section as follow: "The atmospheric CO inversion system benefits from the short atmospheric lifetime of CO and the reliability of satellite CO column retrieval. The CO inversion system was previously evaluated, demonstrating a substantial improvement in CO concentration simulation compared to independent CO observations. Regarding CH₄/CO ERs, evaluation against aircraft measurements revealed a close agreement between field-measured values and the data employed herein. Although uncertainties persist, stemming from the

inversion process, limited spatiotemporal coverage of the evaluation datasets (e.g., the FIREX-AQ and ATom campaigns were only conducted in summer and winter, respectively) and lack of peat fire plume observations. Despite these uncertainties, the lower bound of the uncertainty range (averaging 18.1 Tg yr⁻¹ for 2003–2020) remains in close alignment with the average estimate from four global fire emission models (18.9 Tg yr⁻¹ for the same period), suggesting that they are unlikely to significantly alter the study's main conclusions. "

4. A central conclusion of this study is that the discrepancy between the CO inversion + CH₄/CO ratio CH₄ emissions and prior wildfire emissions are due to undetected small fires, as the title of the manuscript states. The authors justify this based on the high bias of the results compared to previous fire CH₄ estimates, and then use: (1) FIREX-AQ and ATom field campaign observations to show the CH₄/CO ratios are likely not overestimated and use (2): a comparison to high resolution FireCCISFD11 burned areas maps in sub-Saharan Africa to show that coarse resolution datasets are likely causing underestimations. I appreciate the well-reasoned argument here which can convince the reader the conclusion is likely to be true, but the evidence is not sufficient for the conclusion to be certain. The language in the manuscript should reflect this, and perhaps the title as well because the title reads with high certainty.

Response:

Thanks for pointing this out. While we have made every effort to analyze and support the role of undetected small fires, we acknowledge that the evidence is not conclusive and does not permit drawing this conclusion with absolute certainty. In response, we have revised some sentences in the manuscript to reflect this more cautiously, ensuring that the conclusion acknowledges the remaining uncertainties:

"Line 34 –The possible underestimation of previous model results was likely attributed to undetected small fires and underestimated fuel consumption by coarse-resolution data products."

"Line 266 –Our study findings suggest that existing fire emission models may be

underestimating global fire CH₄ emissions due to their reliance on coarse-resolution burned area and fuel consumption data."

We have also adjusted the title to “*Enhanced CH₄ emissions from global wildfires likely due to undetected small fires*” to better align with the evidence presented and to avoid conveying unwarranted certainty. Additionally, in the ***Discussion and implication*** section, we acknowledge the remaining uncertainties, including those related to CO inversion, as you pointed out. Consistent with your suggestion, we also emphasize that further evidence is needed to confirm the primary drivers of the observed "underestimation."

5. L52: Do you mean ‘main’ source of pyrogenic CH₄ emissions

Response:

We have revised the sentence as bellow: "*Global CH₄ sources are associated with biogenic, thermogenic, or pyrogenic processes, with fires being the primary source of pyrogenic CH₄ emissions.*"

6. L61: The range (min – max estimates) is more useful to communicate than the percent difference

Response:

As suggested, we have revised the sentence as bellow: "*Intercomparison studies have demonstrated a range of 6.4–13.2 (min–max) Pg CO₂ yr⁻¹ in global fire emissions across different models²⁶, ...*"

7. L80: “Atmospheric inversion” can be easily confused with the meteorological phenomena of a temperature inversion. It would be better to refer to this as “inverse modelling”

Response:

As suggested, we have revised the sentence as bellow: "*Inverse modeling provides a promising approach to infer CH₄ fluxes from ambient CH₄ observations, ...*"

8. L82: High background CH₄ levels are ubiquitous in the atmosphere, so spatial overlap is

redundant here. This entire sentence can be simplified since the three clauses are essentially referring to the same problem.

Response:

As suggested, we have revised the sentence as bellow: "*However, distinguishing fire CH₄ emissions from total CH₄ fluxes is challenging due to contributions from other human and natural sources and interactions among multiple CH₄ sources.*"

9. L97: This sentence is confusing because it sounds like the CH₄ emissions are computed in the modelling setup (i.e. a multi-species inversion), when in reality the CO emissions are being postprocessed with conversion factors.

Response:

As suggested, we have revised the sentence as bellow: "*..., and applied these functions post-inversion to the CO emissions derived from our inversion system.*"

10. L107: Again, it does not seem correct to refer to the results as “satellite-observed CO inversion-based” estimates, given how dependent the outcomes are on the second step of converting optimized CO emissions to CH₄ emissions.

Response:

Thank you for your suggestion, which aligns closely with your second comment. We have revised the relevant statements throughout the manuscript accordingly. For further details, please refer to our response to the second comment.

11. L121: What does the comparison look like with FINN v2.5?

Response:

We compared our post-CH₄ emission estimates with FINNv2.5 (see figure below) and observed that the discrepancies were most pronounced between 2003–2010, during which FINNv2.5 emissions exceeded our post-CH₄ estimates by approximately 5.2 Tg yr⁻¹. In contrast, from 2011–2020, the differences were smaller, with FINNv2.5 exceeding post-CH₄

estimates by about 1.5 Tg yr⁻¹. Spatially, our post-CH₄ emissions are lower than FINNv2.5 primarily in regions with higher biomass, such as the Amazon Basin, Central Africa, and parts of South Asia, while in most other regions, our estimates are higher than FINNv2.5. Wiedinmyer et al.²⁷ also highlighted significant overestimation by FINNv2.5 in the Amazon Basin and Central Africa when comparing CO simulations with satellite observations, further suggesting overestimation in these regions. To provide readers with a clearer comparison, we have added the following sentences to the *Global fire CH₄ emissions inferred from CO inversion* section: "Compared to our fire CH₄ estimates, FINNv2.5 is generally higher (Supplementary Fig. 1(a)), particularly from 2003–2010, though the estimates converge more closely from 2011–2020 (with an average difference of ~1.5 Tg yr⁻¹). Spatially, our fire CH₄ estimates are higher than those from FINNv2.5 in most regions (Supplementary Fig. 1(b)), except in high-biomass areas such as the Amazon Basin and Central Africa, where Wiedinmyer et al.²⁷ suggested that FINNv2.5 likely overestimates emissions, as indicated by comparisons between model results and satellite observations."

Supplementary Fig. 1: Comparison between global CO-based fire CH₄ emission estimates and FINNv2.5 results. (a) Annual trends in fire CH₄ emissions from CO-based estimates (red curve) and estimates of FINNv2.5 global fire emission model (green curve) from 2003 to 2020, including the fitted linear trends (dashed red and green lines). The shaded gray region represents the range of error

resulting from uncertainties in CH₄/CO emission ratios and inversion-based CO estimates (Methods). Trend assessments are conducted using the nonparametric Mann–Kendall test and Theil–Sen estimator, with 2003–2020 trends and uncertainties provided. Significant trends are denoted by asterisks ($p < 0.1$ and ** $p < 0.05$). (b) Spatial distribution of differences between CO–based CH₄ emission estimates and those of FINVv2.5. Data averaged between 2003 and 2020 are at the spatial resolution of 3.75° longitude × 1.9° latitude.*

12. L175: There should be an attempt to quantify these potential uncertainties.

Response:

Thank you for your suggestion, which aligns closely with your third comment. We have added the discussions of uncertainties for posterior CO emission. For further details, please refer to our response to the third comment.

13. L206: It is difficult to conclude this confidently when you haven't tested the CH₄/CO ERs extensively over every biome.

Response:

As suggested, we have revised the corresponding sentence (Line 218) as follow: "*This alignment with previous aircraft measurements suggested that the uncertainties in CH₄/CO ER were not the dominant factor driving the discrepancies between model results, though further biome-specific testing is required to confirm this across all regions.*"

14. L243: It is better to use “may be underestimating....due to....”

Response:

As suggested, we have revised the corresponding sentence (Line 266) as follow: "*Our study findings suggest that existing fire emission models may be underestimating global fire CH₄ emissions due to their reliance on coarse-resolution burned area and fuel consumption data.*"

15. L256: Please see the general comments on this paragraph.

Response:

We have added the discussions of uncertainties for posterior CO emission. For further details, please refer to our response to the third comment.

16. L273: This is a confusing metric, are you quantifying the difference between your result and the 4 models masked for each country, and then dividing by that country's total anthropogenic emissions?

Response:

To clarify this point, we have revised the sentence (Line 302) as follow: "*The extent of such underestimation, based on the total difference (equivalent to 5.1 Tg yr^{-1}) between our results and the four models, corresponds to a significant proportion, ranging from 8% to 78%, of the total anthropogenic CH_4 emissions (all sectors in the EDGARv7.0 database) for the top 10 emitting countries.*"

17. L437: The full name of MOPITT should be given at least once.

Response:

As suggested, we have included the full name of *Measurements Of Pollution In The Troposphere (MOPITT)*.

18. L515: How long was the transport time?

Response:

The transport time, referred to as plume age, was determined based on the time since the most recent fire influence, using back trajectories from ATom datasets. This is labeled as 'Fire inf' in Supplementary Table 2. To clarify this we have revised corresponding sentence as follow: "*Plume age was determined based on time since the most recent fire influence, which was based on back trajectories²⁸ obtained from ATom datasets²⁹, as indicated by Fire inf in Supplementary Table 2.*"

19. Figure 1a: The shaded region indicating the uncertainties in CH_4/CO emission ratios

appears to be a constant relative error, is this the case? This needs to be explained more clearly.

Response:

Thanks for pointing out the need for clarification. Please see the details in our response on your third comments about the re-evaluated process for shaded region in Figure 1a. While the CH₄/CO uncertainty itself is constant, the shaded region varies with changes in GFED dry matter and our posterior CO estimates, as the ER in Equation (1) is recalculated based on CH₄/CO uncertainties. To clarify this, we have revised the caption of Fig. 1 as follows: "*The shaded gray region represents the range of error resulting derived from uncertainties in CH₄/CO emission ratios and inversion-based CO estimates, which vary with changes in dry matter and CO estimates (Methods).*"

- 20. Figure 2a: I find the gray dashed differences line to be distracting and redundant with the red and green curves already present.

Response:

We have redrawn Figure 2a, replacing the dashed differences line with an area chart, as shown below:

- 21. Supplementary Table 1: The CO inversion-based estimates should have an error interval associated as they are presented in Supplementary Table 3 for the EFs.

Response:

As suggested, we have included the error interval in Supplementary Table 1.

22. Supplementary Figure 11: The caption for this figure is wordy and difficult to understand. It may help with interpretation to have the anthropogenic CH₄ emissions as well as the fire CH₄ emissions on top of the bars, followed by the percentages to follow the calculations better.

Response:

As with our response to your 16th comment, the proportion shown in the figure is not the difference between our result and the four models masked for each country, divided by that country's total anthropogenic emissions. Instead, it represents the ratio of the total difference (5.1 Tg yr⁻¹) between our results and the four models to the total anthropogenic CH₄ emissions for each country. To clarify, we have revised the relevant captions as follows: "*The total difference (equivalent to 5.1 Tg yr⁻¹) between our results and the four models as a proportion of each country's total anthropogenic CH₄ emissions is indicated by the black numbers on top of the bars.*"

We appreciate the careful and thoughtful appraisal of our work by the Editors/Reviewers, along with their many helpful suggestions. We hope that the corrections made in response to your feedback will meet with your approval.

References

- 1 Duncan, B. N. *et al.* Global budget of CO, 1988–1997: Source estimates and validation with a global model. *Journal of Geophysical Research: Atmospheres* **112** (2007). <https://doi.org/https://doi.org/10.1029/2007JD008459>
- 2 Stein, O. *et al.* On the wintertime low bias of Northern Hemisphere carbon monoxide found in global model simulations. *Atmos. Chem. Phys.* **14**, 9295-9316 (2014). <https://doi.org/10.5194/acp-14-9295-2014>
- 3 Pison I, Bousquet P, Chevallier F, Szopa S, Hauglustaine D. Multi-species inversion of CH₄, CO and H₂ emissions from surface measurements. *Atmos Chem Phys* 2009, **9**(14): 5281-5297. <https://doi.org/10.5194/acp-9-5281-2009>
- 4 Chevallier, F. *et al.* Inferring CO₂ sources and sinks from satellite observations: Method and application to TOVS data. *Journal of Geophysical Research: Atmospheres* **110** (2005). <https://doi.org/https://doi.org/10.1029/2005JD006390>
- 5 Chevallier, F. *et al.* African CO emissions between years 2000 and 2006 as estimated from MOPITT observations. *Biogeosciences* **6**, 103-111 (2009). <https://doi.org/10.5194/bg-6-103-2009>
- 6 Ni, Z.-z. *et al.* Assessment of winter air pollution episodes using long-range transport modeling in Hangzhou, China, during World Internet Conference, 2015. *Environmental Pollution* **236**, 550-561 (2018). <https://doi.org/https://doi.org/10.1016/j.envpol.2018.01.069>
- 7 Zheng, B., Chevallier, F., Ciais, P., Yin, Y. & Wang, Y. On the Role of the Flaming to Smoldering Transition in the Seasonal Cycle of African Fire Emissions. *Geophysical Research Letters* **45**, 11,998-912,007 (2018). <https://doi.org/https://doi.org/10.1029/2018GL079092>
- 8 Zheng, B. *et al.* Global atmospheric carbon monoxide budget 2000–2017 inferred from multi-species atmospheric inversions. *Earth Syst. Sci. Data* **11**, 1411-1436 (2019). <https://doi.org/10.5194/essd-11-1411-2019>
- 9 Lutsch, E. *et al.* Detection and attribution of wildfire pollution in the Arctic and northern midlatitudes using a network of Fourier-transform infrared spectrometers and GEOS-Chem. *Atmos. Chem. Phys.* **20**, 12813-12851 (2020). <https://doi.org/10.5194/acp-20-12813-2020>
- 10 Worden, J. R. *et al.* Reduced biomass burning emissions reconcile conflicting estimates of the post-2006 atmospheric methane budget. *Nature Communications* **8**, 2227 (2017). <https://doi.org/10.1038/s41467-017-02246-0>
- 11 Chen, Y. *et al.* Multi-decadal trends and variability in burned area from the 5th version of the Global Fire Emissions Database (GFED5). *Earth Syst. Sci. Data Discuss.* **2023**, 1-52 (2023). <https://doi.org/10.5194/essd-2023-182>
- 12 Barker, P. A. *et al.* Airborne measurements of fire emission factors for African biomass burning sampled during the MOYA campaign. *Atmos. Chem. Phys.* **20**, 15443-15459 (2020). <https://doi.org/10.5194/acp-20-15443-2020>
- 13 Zellweger, C. *et al.* Inter-comparison of four different carbon monoxide measurement techniques and evaluation of the long-term carbon monoxide time series of Jungfraujoch. *Atmos. Chem. Phys.* **9**, 3491-3503 (2009). <https://doi.org/10.5194/acp-9-3491-2009>

-
- 14 Lowry, D. *et al.* Marked long-term decline in ambient CO mixing ratio in SE England, 1997–2014: evidence of policy success in improving air quality. *Scientific Reports* **6**, 25661 (2016). <https://doi.org/10.1038/srep25661>
- 15 Parrish, D. D. *et al.* Decadal change in carbon monoxide to nitrogen oxide ratio in U.S. vehicular emissions. *Journal of Geophysical Research: Atmospheres* **107**, ACH 5-1-ACH 5-9 (2002). <https://doi.org/https://doi.org/10.1029/2001JD000720>
- 16 McDuffie, E. E. *et al.* A global anthropogenic emission inventory of atmospheric pollutants from sector- and fuel-specific sources (1970–2017): an application of the Community Emissions Data System (CEDS). *Earth Syst. Sci. Data* **12**, 3413-3442 (2020). <https://doi.org/10.5194/essd-12-3413-2020>
- 17 Crippa, M. *et al.* Gridded emissions of air pollutants for the period 1970–2012 within EDGAR v4.3.2. *Earth Syst. Sci. Data* **10**, 1987-2013 (2018). <https://doi.org/10.5194/essd-10-1987-2018>
- 18 Andela, N. *et al.* A human-driven decline in global burned area. *Science* **356**, 1356-1362 (2017). <https://doi.org/10.1126/science.aal4108>
- 19 Nisbet, E. G. *et al.* Methane Mitigation: Methods to Reduce Emissions, on the Path to the Paris Agreement. *Reviews of Geophysics* **58**, e2019RG000675 (2020). <https://doi.org/https://doi.org/10.1029/2019RG000675>
- 20 van der Werf, G. R. *et al.* Global fire emissions estimates during 1997–2016. *Earth Syst. Sci. Data* **9**, 697-720 (2017). <https://doi.org/10.5194/essd-9-697-2017>
- 21 Hall, J. V. *et al.* GloCAB: global cropland burned area from mid-2002 to 2020. *Earth Syst. Sci. Data* **16**, 867-885 (2024). <https://doi.org/10.5194/essd-16-867-2024>
- 22 Lan, R., Eastham, S. D., Liu, T., Norford, L. K. & Barrett, S. R. H. Air quality impacts of crop residue burning in India and mitigation alternatives. *Nature Communications* **13**, 6537 (2022). <https://doi.org/10.1038/s41467-022-34093-z>
- 23 Neef, L., van Weele, M. & van Velthoven, P. Optimal estimation of the present-day global methane budget. *Global Biogeochemical Cycles* **24** (2010). <https://doi.org/https://doi.org/10.1029/2009GB003661>
- 24 MOYA/ZWAMPS Team. *et al.* Isotopic signatures of methane emissions from tropical fires, agriculture and wetlands: the MOYA and ZWAMPS flights. *Philosophical Transactions of the Royal Society A: Mathematical, Physical and Engineering Sciences* **380**, 20210112 (2021). <https://doi.org/10.1098/rsta.2021.0112>
- 25 Crippa, M., Guizzardi, D., Solazzo, E., Muntean, M., Schaaf, E., Monforti-Ferrario, F., Banja, M., Olivier, J.G.J., Grassi, G., Rossi, S., Vignati, E., GHG emissions of all world countries - 2021 Report, EUR 30831 EN, Publications Office of the European Union, Luxembourg, 2021, ISBN 978-92-76-41547-3, [doi:10.2760/173513](https://doi.org/10.2760/173513), JRC126363.
- 26 Liu, T. *et al.* Diagnosing spatial biases and uncertainties in global fire emissions inventories: Indonesia as regional case study. *Remote Sensing of Environment* **237**, 111557 (2020). <https://doi.org/https://doi.org/10.1016/j.rse.2019.111557>
- 27 Wiedinmyer, C. *et al.* The Fire Inventory from NCAR version 2.5: an updated global fire emissions model for climate and chemistry applications. *Geosci. Model Dev.* **16**, 3873-3891 (2023). <https://doi.org/10.5194/gmd-16-3873-2023>

28 Chen, X. *et al.* HCOOH in the Remote Atmosphere: Constraints from Atmospheric Tomography (ATom) Airborne Observations. *ACS Earth and Space Chemistry* **5**, 1436-1454 (2021). <https://doi.org/10.1021/acsearthspacechem.1c00049>

29 Ray EA. ATom: Back Trajectories and Influences of Air Parcels Along Flight Track, 2016-2018. ORNL Distributed Active Archive Center; 2022.

Responses to reviewers' comments

We appreciate the reviewer's careful and thoughtful comments on our manuscript entitled "*Enhanced CH₄ emissions from global wildfires likely due to undetected small fires*" and thank the helpful suggestions to improve the article. We have carefully reviewed all comments and revised the article accordingly. The sentences are depicted in red in the text to highlight the new addition and strikethroughs are used to show deletion. All the responses are in the green background below.

Responses to Reviewer 2 comments:

1. This is an interesting and potentially important paper. The authors have responded well to my comments, though I still have some minor quibbles (see below). I also remain concerned that the authors' image of moist tropical Africa as a quasi-natural land surface environment, not the reality of widespread intense farming, many huge new cities, and abundant cars, trucks and small smouldering fires. That said, the pepr should now be accepted and should go forwards to publication

Response:

We sincerely thank you for your thoughtful feedback and positive evaluation of our manuscript. Regarding your concern about the representation of moist tropical Africa as a quasi-natural land surface environment, we acknowledge the importance of considering the extensive human influence in this region, including intense farming, urbanization, and emissions from vehicles and small smoldering fires. While our study primarily focuses on wildfire emissions, we recognize that anthropogenic factors play a critical role in shaping the emission landscape. To address this, we have revised the first paragraph of the discussion section to better acknowledge these anthropogenic contributions and their potential overlap with wildfire emissions:

"Although these observations were only based on a regional analysis of Africa, this continent accounts for more than half of the global burned area and fire emissions. We acknowledge that tropical Africa is not a quasi-natural land surface, as it is heavily impacted by various emissions from anthropogenic activities, which could affect emission patterns and their representation in fire-related datasets. High-resolution data from different regions, especially those that consider anthropogenic activities, will aid in a more comprehensive evaluation."

2. Line 58 on says "Moreover, CH₄ emissions generated by fires are isotopically heavier than those of biogenic and thermogenic origin, with C3 vegetation (e.g., trees) producing lighter isotopes and C4 vegetation (e.g., grasses) producing heavier ones."

Two comments

1. Delete 'and thermogenic' – 'thermogenic' means 'fire-generated' (although here used in the sense of geological emissions. Moreover many geological (thermogenic) methane emissions are isotopically heavy, and some geological emissions (e.g. from deep coal mines) can even be heavier than fire emissions (particularly those from burning C3 plants)

2. Poor language – 'heavier ones' What does that mean?

Maybe change these lines to something like

"Moreover, CH₄ emissions generated by fires are isotopically heavier than methane of biogenic origin, as methane from burning C3 vegetation (e.g. trees), and especially from burning C4 vegetation (e.g. many tropical grasses, maize, sugar cane), is typically enriched in ¹³C compared to biogenic emissions from wetlands, ruminants, or waste.

Response:

Thank you for your valuable suggestion. We have deleted the term “and thermogenic” and revised the corresponding sentences as follows:

"Moreover, CH₄ emissions generated by fires are isotopically heavier than methane of biogenic origin¹, as methane from burning C3 vegetation (e.g., trees), and especially from burning C4 vegetation (e.g., many tropical grasses, maize, sugar cane)², is typically enriched in ¹³C compared to biogenic emissions from wetlands, ruminants, or waste."

3. Line 228 has a spelling error 'emissioin'

Response:

We have corrected "emissioin" to "emission".

Responses to Reviewer 3 comments:

1. Thank you to the authors for providing revisions on the manuscript "Enhanced CH₄ emissions from global wildfires likely due to undetected small fires", including a revision of the title to better represent the study conclusions. My concerns on 1) the inclusion of an uncertainty estimate for the inverse-modelling results, and 2) the scientific communication with respect to the methodology, results, and conclusions of this modelling study have been addressed. I am happy to recommend this manuscript for publication.

Response:

Thank you very much for your positive feedback. We are pleased to hear that our revisions have addressed your concerns regarding the uncertainty estimates and the scientific communication of the methodology, results, and conclusions. We appreciate your recommendation for publication and are grateful for the opportunity to improve the manuscript.

References

1. Neef, L., van Weele, M. & van Velthoven, P. Optimal estimation of the present-day global methane budget. *Global Biogeochemical Cycles* **24** (2010).
<https://doi.org/https://doi.org/10.1029/2009GB003661>
2. MOYA. *et al.* Isotopic signatures of methane emissions from tropical fires, agriculture and wetlands: the MOYA and ZWAMPS flights. *Philosophical Transactions of the Royal Society A: Mathematical, Physical and Engineering Sciences* **380**, 20210112 (2021).
<https://doi.org/10.1098/rsta.2021.0112>